# Towards General Continuous Memory for Vision-Language Models

**Wenyi Wu**[*], **Zixuan Song**[*], **Kun Zhou**[†], **Yifei Shao, Zhiting Hu, Biwei Huang**

University of California, San Diego.

kuzhou@ucsd.edu

## Abstract

Language models (LMs) and their extension, vision-language models (VLMs), have achieved remarkable performance across various tasks. However, they still struggle with complex reasoning tasks that require multimodal or multilingual real-world knowledge. To support such capabilities, an external memory system that can efficiently provide relevant multimodal information is essential. Existing approaches generally concatenate image and text tokens into a long sequence as memory, which, however, may drastically increase context length and even degrade performance. In contrast, we propose using continuous memory-a compact set of dense embeddings-to more effectively and efficiently represent multimodal and multilingual knowledge. Our key insight is that a VLM can serve as its own continuous memory encoder. We empirically show that this design improves performance on complex multimodal reasoning tasks. Building on this, we introduce a data-efficient and parameter-efficient method to fine-tune the VLM into a memory encoder, requiring only 1.2% of the model's parameters and a small corpus of 15.6K self-synthesized samples. Our approach CoMEM utilizes VLM's original capabilities to encode arbitrary multimodal and multilingual knowledge into just 8 continuous embeddings. Since the inference-time VLM remains frozen, our memory module is plug-and-play and can be flexibly integrated as needed. Extensive experiments across eight multimodal reasoning benchmarks demonstrate the effectiveness of our approach. Code and data is publicly released here https://github.com/WenyiWUO111/CoMEM.

## 1 Introduction

Through large-scale training, language models (LMs) [1, 2] have demonstrated remarkable performance across diverse real-world tasks. LMs even surpass human capabilities in language reasoning tasks [3] such as mathematical problem solving [4], commonsense reasoning[5], and code synthesis [6]. However, when confronted with complex reasoning tasks that demand multimodal or multilingual world knowledge, both LMs and their

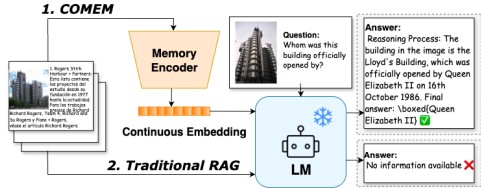

Figure 1: CoMEM architecture in comparison to the traditional RAG method.

vision-language model (VLM) extensions continue to face significant challenges [7, 8], primarily due to insufficient world knowledge representation.

---

[*]Equal Contribution

[†]Corresponding Author

39th Conference on Neural Information Processing Systems (NeurIPS 2025).

Inspired by how humans offload facts, plans, and ideas to external repositories like notebooks or databases for on-demand access, it is promising to develop a general external memory[3] that contains useful world knowledge for augmenting VLMs [9, 10]. Early approaches directly concatenate the collected useful information into a long token sequence, and feeds it into VLMs [9, 11] *e.g.,* retrieval-augmented generation (RAG) methods. However, multimodal representations demand significantly more input tokens (*e.g.,* 8 to 11427 tokens per image in Qwen2.5-VL [12]). Thus, simple concatenation would greatly increase the input length, making it difficult for the memory content to be used [13] (see the degradation in performance shown in Table 2 after using RAG). To solve the token overload issue, token pruning methods have been proposed to remove unimportant in-context tokens [14, 15]. However, token pruning generally leads to incomplete contextual contents, which impedes the VLM's ability to accurately understand and utilize the compressed information [8].

Compared to discrete tokens, continuous embeddings naturally have stronger representation capability for complex data [16, 17, 18]. This advantage makes them particularly promising for memory encoder architectures designed to condense multimodal information into continuous representations. However, training such encoders faces two key challenges: (1) achieving generalizable compression ability across diverse multimodal inputs, and (2) maintaining semantic alignment with the VLM [19]. While large-scale training can improve performance, it greatly increases the training cost and becomes heavily sensitive to the training data distribution. For example, when dominated by simple cases or a single domain, the encoder tends to overfit and degrade generalization performance [20] [21].

In this paper, we focus on efficiently training a general continuous memory encoder to effectively supply multimodal knowledge for VLMs. To avoid costly training for semantic alignment, it is essential to minimize the representation gap between the memory encoder and downstream VLMs before training. Therefore, a natural way is to use the VLM itself as the memory encoder. Our empirical study confirms that the VLM can serve as a memory encoder for itself without any additional training. Benefiting from the stacked self-attention mechanism, its generated continuous embeddings in each layer have already aggregated rich semantic information [22] [23]. As shown in Fig. 2, even a simple rule-based embedding selection strategy for constructing the memory can greatly boost the performance of VLMs in complex multimodal multilingual reasoning tasks, compared to RAGs.

Based on our empirical findings, we propose a data-efficient and parameter-efficient training recipe to further improve the compressor rate and adaptation performance of the VLM-based general continuous memory encoder. Concretely, we only need to fine-tune the low-rank adaptation matrices (LoRA) [24] in the VLM-based memory encoder, and a lightweight Q-Former [25] for further compressing the VLM representations into only eight embeddings, 1.2% parameters in total. In terms of data, we only need the VLM itself to synthesize 15.6k samples for training. This efficient training strategy enables our continuous memory to reuse the original ability of the VLM, to effectively encode multimodal and multilingual knowledge. Since we do not need to train the inference VLM, our memory is plug-and-play and can be flexibly integrated with the VLM when necessary.

To demonstrate the effectiveness of our approach, we apply our method to state-of-the-art VLMs, and evaluate the performance across eight visual reasoning benchmarks. For six English visual reasoning benchmarks, our method achieves an average improvement of +8.0% on Qwen2-Instruct-VL and +7.7% on Qwen2.5-Instruct-VL. On two multilingual multimodal benchmarks, our approach further improves performance by +5.1% and +4.3% on Qwen2-Instruct-VL and Qwen2.5-Instruct-VL, respectively. Furthermore, our adaptation study results also indicate the transferability of our VLM-based memory encoder to improve LMs in visual reasoning tasks. The long context understanding study also exhibits the stable and superior performance of our method.

## 2 Empirical Analysis with VLM as Memory Encoder

In this section, we conduct an empirical study to examine (1) whether VLM can serve as a continuous memory encoder to compress multimodal information into compatible embeddings, and (2) whether a few embeddings from the VLM can preserve key information to improve multimodal reasoning tasks.

---

[3]In this paper, external memory denotes any detachable module or function that supplies the knowledge without changing LM parameters, in contrast to internal memory that embeds knowledge by modifying parameters.

Table 1: Comparison between our approach and other representative line of work.

| Category | Method | Properties | | Scenarios | | Training Cost | |
|---|---|---|---|---|---|---|---|
| | | Continuous | Pulg-and-Play | Multimodal | Multilingual | Data | Parameters |
| Multimodal-RAG | EchoSight [26] | ✗ | ✓ | ✗Text | ✗ | 900K | 300M |
| | ReflectiVA [27] | ✗ | ✗ | ✓Image+Text | ✗ | 6.82M | 8B |
| | RoRA-VLM [28] | ✗ | ✗ | ✓Image+Text | ✗ | 1M | 7B |
| Context-Compression | xRAG [29] | ✓ | ✓ | ✗Text | ✗ | 3M | 40M |
| | KV-Distill [30] | ✗ | ✓ | ✗Text | ✗ | 500K+ | 150M |
| | VoCo-LLaMA [31] | ✓ | ✗ | ✓Image/Video | ✗ | 665K | 7B |
| LM Memory | LONGMEM [10] | ✓ | ✓ | ✗Text | ✗ | 114M | 558M |
| | MA-LMM [32] | ✓ | ✓ | ✓Video+Text | ✗ | NA | 200M |
| | M+ [33] | ✓ | ✗ | ✗Text | ✗ | 5M | NA |
| | MemGPT [34] | ✗ | ✓ | ✗Text | ✗ | NA | NA |
| | CoMEM | ✓ | ✓ | ✓Image+Text | ✓ | 15K | 200M |

## 2.1 Analysis Setup

For the empirical study, we conduct experiments on two state-of-the-art VLMs, *i.e.,* Qwen2-VL-7B and Qwen2.5-VL-7B, and test the performance on three multimodal reasoning benchmarks.

**Evaluation Settings.** To compare the effectiveness of different memory and context compression methods, we select three benchmarks: InfoSeek [35], OK-VQA [36], and A-OKVQA [37]. These benchmarks contain complex visual questions that require both accurate visual entity identification and multi-step reasoning to derive the correct answer. Following existing work [38] [28], for each question, we utilize CLIP-based retriever [39] to collect relevant top-10 multimodal knowledge items from a Wikipedia-based source dataset WiT [40] to construct the input data for the memory.

Table 2: Comparison of training-free memory methods. Bold indicates the best performance. For VLM-as-Memory here, we use the cache KV from a VLM without fine-tuning, which differs from the main method described in Section 3 and is intended for preliminary exploration.

| Backbone Model | Method | InfoSeek | | | OKVQA | AOKVQA |
|---|---|---|---|---|---|---|
| | | Query | Entity | All | | |
| Qwen2.5-VL-Instruct | - | 22.5 | 22.4 | 22.5 | 35.0 | 39.8 |
| | +RAG | 17.7 | 18.8 | 18.2 | 31.3 | 34.9 |
| | +FastV | 26.2 | 22.6 | 24.2 | 31.5 | 34.9 |
| | +VLM-as-Memory | 29.3 | **28.0** | **28.6** | 37.3 | **44.4** |
| | +VLM-as-Memory+AS | **30.0** | 25.3 | 27.5 | **37.9** | 41.8 |
| Qwen2-VL-Instruct | - | 17.9 | 17.8 | 17.9 | 36.3 | 41.8 |
| | +RAG | 22.7 | 19.0 | 20.5 | 41.9 | 45.3 |
| | +FastV | 23.6 | 23.8 | 23.7 | **42.0** | **45.4** |
| | +VLM-as-Memory | 28.8 | **29.7** | 29.3 | 37.7 | 38.9 |
| | +VLM-as-Memory+AS | **31.7** | 28.8 | **30.2** | 34.3 | 36.4 |

**Memory Methods.** We test the effectiveness of our VLM-as-memory method by comparing with RAG, token pruning, and our variations using different embedding selection strategies.

• *Vanilla RAG*: it simply concatenates all the multimodal knowledge items into a long sequence, and then feeds it with the visual question as the input of VLM.

• *FastV* [14]: it adopts a token pruning strategy that discard the image tokens with lower attention scores from the multimodal knowledge items. Then, the pruned token sequence is fed into the VLM.

• *VLM-as-Memory*: we utilize the VLM itself to encode the knowledge items, and extract the hidden states in all layers as the memory. These are concatenated at corresponding layers of the VLM. For efficiency, we only add the memory in 17-19 layers.

• *VLM-as-Memory+Attn*: we utilize the average attention scores across all layers within the VLM to select the top-25% key continuous embeddings to compose the memory.

## 2.2 Results and Findings

In this part, we present the results and discuss the findings to analyze whether VLM can be a memory encoder, assessing both their compression efficiency and semantic alignment capabilities.

**Effectiveness Study of VLM-as-Memory Methods.** As shown in Table 2, the vanilla RAG method causes performance degradation in several tasks, underscoring its limitations for memory integration. FastV mitigates this issue by pruning redundant tokens in the memory, resulting in measurable improvements. Notably, the VLM-as-Memory method outperforms both approaches across most tasks, suggesting that the VLM's continuous embeddings are inherently more compatible with its own processing than token-based input. Furthermore, with the addition of a simple attention-based compression mechanism, the VLM-as-Memory method achieves even greater performance gains. Thus, we conclude that:

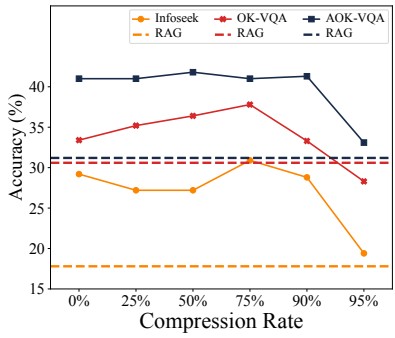

Figure 2: Qwen2.5 accuracy with varying attention-based compression rates.

*(1) VLMs can effectively serve as their own memory encoders for external multimodal knowledge.* The continuous embeddings they generate can be directly reused by the same model without requiring additional training.

*(2) The continuous embeddings produced by VLMs effectively preserve knowledge content.* They remain robust under simple compression strategies and reliably enhance performance.

**Compressibility Study of VLM-as-Memory Methods.** To study the compressibility, another key feature of an effective memory mechanism, we investigate how performance varies under different compression rates using the VLM-as-Memory approach. As shown in Fig. 2, despite employing a simple token selection strategy, our method outperforms the baseline even at a high compression rate of 5This suggests that a small number of continuous embeddings already encapsulate most of the essential contextual information from the input. Therefore, we can conclude that:

*(3) The continuous embeddings produced by VLMs support high compression rates.* This highlights the potential for achieving even greater compression through more advanced compression methods.

## 3 Approach

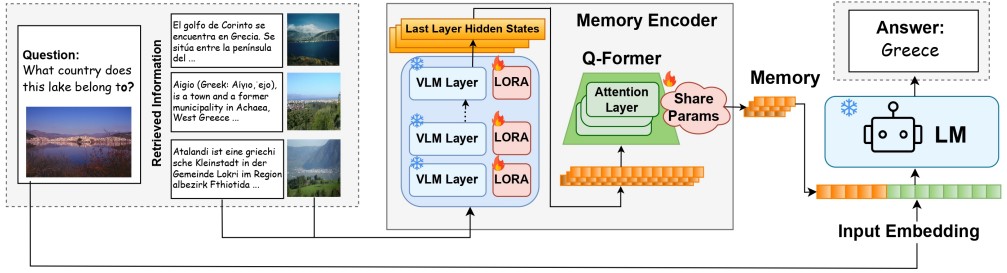

Figure 3: Overview of the CoMEM architecture. Given a vision-language query, the system retrieves relevant multimodal knowledge via visual features. Retrieved image-text pairs are processed by a Memory Encoder—which consists of a VLM and Q-Former—to generate a dense continuous memory. This memory and the original query are fed into a frozen LM to produce accurate, grounded answers.

According to our empirical study, the VLM can be an effective memory encoder for itself, owing to the satisfactory semantic alignment and compressibility of its produced continuous embeddings.

Building on this insight, we aim to efficiently train the VLM into a continuous memory encoder, to supply supplementary multimodal knowledge during inference. Concretely, we add a trainable lightweight Q-Former to control the compression rate, synthesize a small training dataset using the VLM itself, and perform data-efficient and parameter-efficient training.

## 3.1 Task Definition.

We aim to train a general-purpose continuous memory encoder capable of mapping arbitrary multimodal and multilingual data into continuous embeddings that augment the knowledge of a VLM. To ensure plug-and-play compatibility, we keep the VLM's parameters frozen during inference, while continuous memory embeddings are directly used for downstream tasks. To achieve this, the memory encoder should (1) efficiently condense diverse multimodal and multilingual data and (2) produce embeddings that are both readable and functional for the VLM.

In this paper, we focus on using general continuous memory to enhance VLMs in complex multimodal reasoning tasks. Formally, given an instance comprising an image $i$ and a natural language question $q$, the task is to predict an accurate answer $a$. Following prior work [28], we assume access to relevant multimodal knowledge items (from external knowledge source), each consisting of an image $\tilde{i}$ and a natural language description $\tilde{d}$. Our memory encoder learns to transform each knowledge item into a continuous vector, formulated as $\mathbf{V}_t = f(\tilde{i}_t, \tilde{d}_t)$. These vectors are aggregated into a unified memory, which the VLM then utilizes for answer prediction: $p(a|i, q, \{\mathbf{V}_t\}_{t=1}^k)$.

## 3.2 VLM-Based Continuous Memory

For our continuous memory, the core idea is to leverage the VLM with a Q-Former as the encoder, and adopt a simple plug-and-play mechanism that enables the VLM to use the memory information.

**Continuous Encoder.** Given each multimodal knowledge item $\langle \tilde{i}_t, \tilde{d}_t \rangle$, we first use the VLM to encode it and collect the continuous representations $\mathbf{E}_t$ in the last layers. Then, we employ a query Transformer (Q-Former) as the compressor to condense $\mathbf{E}_t$ into $k$ continuous embeddings $\mathbf{V}_t$. The Q-Former consists of $k$ query embeddings $\mathbf{q}$ and $L$ Transformer layers. In the first layer, the query embeddings attend to all the continuous representations from the VLM through the cross-attention mechanism. The output representations are then used as query embeddings for the next layer, and the final layer outputs serve as the memory vector $\mathbf{V}_t$. The whole process is formulated as:

$$\mathbf{H}^{(0)} = \mathbf{q}, \qquad \mathbf{H}^{(\ell)} = \text{TransformerLayer}^{(\ell)}\big(\mathbf{H}^{(\ell-1)}, \mathbf{E}_t\big), \quad \mathbf{V}_t = \mathbf{H}^{(L)} \tag{1}$$

To reduce the parameter scale of the Q-Former, we share parameters across all Transformer layers, and set $k = 8$. In this way, only a few parameters are added, and any multimodal knowledge item will be compressed into 8 continuous embeddings. This design ensures lower training cost and a higher compression rate[4], which is helpful to handle large-scale knowledge items and save the storage cost.

**Plug-and-Play Mechanism.** After obtaining the continuous embedding set $\{\mathbf{V}_t\}_{t=1}^n$ for all multimodal knowledge items, we adopt a simple plug-and-play mechanism to equip the VLM the memory. Concretely, we simply concatenate the embeddings into a sequence of $8 \times n$ continuous vectors as the memory, which is prepended to the input embedding $\mathbf{E}_I$ of the VLM during the inference time, formulated as $[\mathbf{V}_1; \cdots; \mathbf{V}_n, \mathbf{E}_I]$. In this way, the VLM can naturally perform autoregressive generation to predict the answer, using its originally learned knowledge and capabilities.

## 3.3 Efficient Training Recipe

Since we introduce the Q-Former, we need to train its parameters to achieve full alignment between the continuous memory and the VLM. Thanks to our design that employs the VLM as the memory encoder, this alignment can be efficiently accomplished through parameter-efficient training using only a small amount of self-synthetic multimodal and multilingual data.

---

[4]Notably, the average token number of knowledge items in this work is **643.7**, and few extremely long ones contain more than 2000 tokens. By compressing into 8 tokens, we can achieve more than **80×** compression rate.

**Training Data Self-synthesis.** To ensure training efficiency, we construct our training dataset by synthesizing responses using the VLM itself, based on multilingual and multimodal questions from existing benchmarks. Specifically, we begin by selecting questions from the training sets of InfoSeek [35], Encyclopedic-VQA (EVQA) [41], and OK-VQA [36] to ensure coverage of diverse multimodal reasoning tasks. For each question, we retrieve three relevant image-text pairs from the WIT [40] knowledge base using CLIP, following the retrieval setup in prior work [28]. These pairs serve as supplementary multimodal knowledge items. We concatenate the question with knowledge items and input the sequence into Qwen2.5-VL-Instruct to simulate a vanilla RAG setting. Only outputs yielding correct answers are retained, resulting in 13.8k high-quality training instances. To extend our dataset beyond English, we randomly select 200 training samples and employ GPT-4o-mini to translate the text part into nine languages: Bulgarian, Chinese, Egyptian Arabic, Filipino, French, Japanese, Portuguese, Russian, and Spanish. This results in an additional 1.8k multimodal multilingual training samples, which aims at activating our model's cross-lingual capabilities. In total, our final fine-tuning corpus for continuous memory includes 15.6K curated samples, covering a variety of multimodal tasks and languages.

**Parameter-efficient Fine-tuning.** Given the above training data, we perform parameter-efficient fine-tuning on the Q-Former and LoRA layers in the VLM encoder. For efficiency, we apply LoRA with a rank of 16 and share parameters across all layers of the Q-Former. Therefore, only 1.2% of total parameters are trainable. The above parameter and data efficient designs guarantee that our entire training process can be completed on a single NVIDIA H100 GPU in 20 hours. We also empirically find the training converges fast, and a single epoch is sufficient to achieve strong performance.

### 3.4 Discussion

In Table 1, we compare our method CoMEM with ten closely related works: *i.e.,* multimodal RAG (EchoSight [26] ReflectiVA [27], and RoRA-VLM [28]), context compression (xRAG [29], KV-Distill [30] and VoCo-LLaMA [31]), and LLM memory methods (LONGMEM [10], MA-LMM [32], M+ [33], and MemGPT [34]). The comparison spans three dimensions: Properties, where we examine whether the method is continuous and plug-and-play; Scenarios, evaluating support for multimodal and multilingual inputs; and Training Cost, which includes the amount of training data required and trainable parameters.

While some existing methods also adopt continuous embeddings and support plug-and-play usage, they often require substantial training resources—typically involving millions of training samples and extensive parameter updates. In contrast, our method achieves comparable functionality with significantly reduced cost: it utilizes only 15.6k self-synthesized training samples and fine-tunes just 200M parameters, amounting to only 1.2% of the full model. Moreover, a key advantage of our method is our method can handle both multimodal (text and image) and multilingual data, which is very helpful for potential applications in low-resource language settings.

In summary, our proposed method, CoMEM, provides a generalizable, scalable, and compute-efficient solution for augmenting VLMs with a continuous memory mechanism. By leveraging the VLM itself as the memory encoder, CoMEM ensures strong semantic alignment between the memory and the model, while supporting seamless plug-and-play integration for diverse downstream tasks. This design enables effective reasoning over complex multimodal and multilingual inputs, offering a unified and efficient alternative to existing approaches that often rely on discrete context inputs, heavy fine-tuning, or multi-stage retrieval pipelines.

## 4 Experiments

### 4.1 Experimental Setup

**Evaluation Settings** We use WIT [40] (Wikipedia-based Image Text Dataset) as our retrieval knowledge base. Building upon this, we conduct experiments across eight multimodal and multilingual reasoning benchmarks, including six multimodal reasoning benchmarks: InfoSeek [35], OVEN [42], MRAG-Bench [43], OK-VQA [36], A-OKVQA [37], and ViQuAE [44], and two multilingual benchmarks: CVQA [45] and multilingual InfoSeek. Here we use GPT-4o-mini [46] to translate the InfoSeek from English into five different languages to match the language settings of CVQA.

Note (1) InfoSeek and OVEN are constructed from Wikipedia and consist of challenging factual questions. (2) MRAG-Bench, OK-VQA, A-OKVQA, and ViQuAE focus on multimodal real-world, knowledge-intensive tasks. (3) CVQA and multilingual InfoSeek evaluate model's ability to reason diverse linguistic and cultural contexts. Further details about benchmarks are in Appendix A.

**Baseline Methods**    We compare our method against three types of baselines: (1) VLMs, (2) VLMs with vanilla RAG, and (3) advanced RAG methods, covering a total of 18 different models.

For VLMs, we evaluate their original capabilities on multimodal reasoning tasks without access to external knowledge, including: LLaVA-v1.5 [47], LLaVA-v1.6 [47], LLaVA-NeXT-LLaMA3 (denoted as LLaMA3 in tables) [48], InternLM-XComposer2.5vl (InternLM2.5vl) [49], mPLUG-Owl3 [50], Qwen2-VL-Instruct (Qwen2-VL) [51], and Qwen2.5-VL-Instruct (Qwen2.5-VL) [12].

For VLMs with vanilla RAG, we directly insert retrieved image-text pairs into the input prompts of models, without making any architectural modifications or applying additional fine-tuning. This setup evaluates the effectiveness of naive retrieval-based augmentation.

Wiki-LLaVA and RORA-VLM use two-stage retrieval to improve knowledge relevance, while ReflectiVA adds reflective tokens for self-filtering. All three fine-tune the inference-time model. In contrast, EchoSight trains a separate Q-Former for retrieval without training the inference model. However, they all rely on discrete context inputs, which limits their ability to handle long contexts.

**Implementation Details**    Our experimental pipeline comprises three phases: *Knowledge Retrieval*, *Knowledge Compression*, and *Answer Generation*. To ensure fairness, we consistently use the top-10 retrieved image-text pairs across all experimental settings. We evaluate our method on Qwen2-Instruct-VL and Qwen2.5-Instruct-VL, demonstrating its strong generalization capability across different VLMs and question types. More implementation details can be found in Appendix B.

## 4.2  Main Results

**Evaluation on Multimodal Reasoning Task**    Table 3 presents the performance comparison across six multimodal reasoning benchmarks, categorized into *Base Models*, *Retrieval-Augmented Baselines*, and our *Continuous Memory* approach. Among base models, Qwen2-VL and Qwen2.5-VL achieve the highest performance across most benchmarks, which is likely due to their extensive multimodal training corpus and strong vision-language alignment. However, standard RAG integration often leads to inefficiencies in processing longer multimodal inputs, resulting in unstable performance that sometimes underperforms base models. To address this issue, advanced RAG models incorporate mechanisms that retrieve and use relevant content more effectively, resulting in improved performance on reasoning tasks. However, as shown in Section 3.4, existing methods still face limitations, such as difficulty in adapting across modalities or a lack of generalizability across diverse task settings.

In comparison, our approach shows significant gains across multimodal reasoning benchmarks, with particularly strong improvements (over 15%) on OKVQA and A-OKVQA versus baselines. These advancements originate from our VLM-based continuous memory architecture, which exhibits both strong adaptability to different VLMs and excellent generalization across diverse tasks. Remarkably, this level of performance requires minimal fine-tuning (1.2% of parameters on 15.6k samples from InfoSeek, OKVQA and EVQA subsets), yet still achieves remarkable improvements on unseen benchmarks such as OVEN and A-OKVQA. This suggests that our method can effectively fuse multimodal long-context knowledge, and generalize effectively to a wide range of downstream tasks.

**Evaluation on Multimodal Multilingual Reasoning Task**    We further evaluate our model's multilingual reasoning capabilities on the multilingual InfoSeek and CVQA benchmarks. As shown in Table 4, standard RAG methods demonstrate reduced effectiveness for non-English questions, potentially due to misalignment between retrieved multilingual content and input queries. In contrast, our memory mechanism encodes and stores transferable semantic representations that preserve core cross-modal and cross-lingual knowledge. This design translates into consistent accuracy improvements across all evaluated languages, achieving absolute gains of 6–12 points on InfoSeek-All scores while simultaneously showing enhanced performance on CVQA metrics. Notably, the model achieves particularly strong performance gains for Bulgarian (18%) and Russian (10%), underscoring the value of our language-agnostic memory mechanism for lower-resource settings where high-quality

Table 3: Performance comparison with three types of baselines on knowledge-intensive VQA benchmarks. Bold indicates the best performance, and underscore denotes the second-best.

| Model | InfoSeek Q | InfoSeek E | OVEN Q | OVEN E | MRAG | OKVQA | AOKVQA | ViQuAE | Avg. |
|---|---|---|---|---|---|---|---|---|---|
| LLaVA-v1.5 | 8.3 | 8.9 | 20.0 | 3.4 | 34.6 | 17.0 | 17.4 | 11.1 | 15.1 |
| LLaVA-v1.6 | 10.3 | 9.1 | 17.9 | 1.8 | 33.4 | 31.4 | 31.7 | 18.7 | 19.3 |
| LLaMA3 | 10.7 | 8.6 | 16.8 | 0.8 | 33.5 | 23.7 | 25.3 | 17.2 | 17.1 |
| InternLM-2.5vl | 13.4 | 10.8 | 14.5 | 3.3 | 34.8 | 29.1 | 32.8 | 29.7 | 19.5 |
| mPLUG-Owl3 | 9.6 | 6.4 | 20.7 | 1.9 | **45.0** | 31.9 | 33.0 | 23.1 | 21.4 |
| Qwen2-VL | 17.9 | 17.8 | 25.5 | 9.3 | 39.3 | 36.3 | 41.8 | 34.5 | 27.8 |
| Qwen2.5-VL | 22.5 | 22.4 | 29.3 | 16.3 | 42.0 | 35.0 | 39.8 | **39.0** | 30.8 |
| LLaVA-v1.5 + RAG | 14.6 | 11.4 | 11.7 | 7.6 | 34.7 | 9.8 | 8.7 | 7.6 | 13.3 |
| LLaVA-v1.6 + RAG | 6.7 | 5.8 | 9.7 | 1.2 | 32.6 | 25.6 | 22.6 | 17.0 | 15.2 |
| LLaMA3 + RAG | 12.1 | 10.8 | 24.7 | _21.5_ | 36.4 | 20.7 | 22.1 | 18.1 | 20.8 |
| InternLM-2.5vl + RAG | 10.5 | 9.5 | 15.2 | 13.6 | 34.3 | 25.9 | 27.8 | 29.6 | 20.8 |
| mPLUG-Owl3 + RAG | 12.6 | 7.2 | 18.0 | 12.0 | 41.9 | 24.7 | 26.4 | 22.5 | 20.7 |
| Qwen2-VL + RAG | 22.7 | 19.0 | 24.7 | _21.5_ | 40.4 | 41.9 | 45.3 | 33.6 | 31.1 |
| Qwen2.5-VL + RAG | 17.7 | 18.8 | 23.0 | 19.7 | _42.1_ | 31.3 | 34.9 | 33.5 | 27.6 |
| Wiki-LLaVA | 28.6 | 25.7 | - | - | - | - | - | - | 27.2 |
| RORA | 27.3 | 25.1 | 26.2 | 15.1 | - | - | - | - | 22.9 |
| EchoSight | 18.0 | 19.8 | - | - | 41.3 | 20.0 | 16.9 | 25.2 | 23.5 |
| ReflectiVA | 28.6 | 28.1 | - | - | 39.7 | 47.5 | 47.6 | 29.8 | _36.9_ |
| **CoMEM + Qwen2VL** | _32.6_ | **33.1** | **30.5** | **23.6** | 35.1 | **57.7** | **60.6** | _36.3_ | **38.7** |
| **CoMEM + Qwen2.5VL** | **32.8** | _28.5_ | _26.0_ | 20.8 | 38.1 | _47.6_ | _55.0_ | 34.7 | 35.4 |

retrieval is hardest to obtain. Overall, the results show that our method enables more robust grounding of multilingual queries and enhances reasoning capabilities across diverse tasks.

Table 4: Performance comparison on Multilingual knowledge-intensive VQA benchmarks.

| Language | Method | Qwen2.5-Instruct-VL Multilingual InfoSeek Unseen-Q | Unseen-E | All | CVQA | Qwen2-Instruct-VL Multilingual InfoSeek Unseen-Q | Unseen-E | All | CVQA |
|---|---|---|---|---|---|---|---|---|---|
| Chinese | - | 17.4 | 13.8 | 15.4 | **82.32** | 15.1 | 10.9 | 12.6 | **74.60** |
| | + RAG | 14.8 | 9.8 | 11.8 | 74.60 | 11.5 | 8.9 | 10.1 | 72.35 |
| | + CoMEM | **22.5** | **21.5** | **22.0** | 78.46 | **23.1** | **19.5** | **21.1** | 73.31 |
| Russian | - | 15.6 | 14.7 | 15.1 | 66.50 | 13.0 | 13.8 | 13.4 | **71.00** |
| | + RAG | 10.6 | 8.9 | 9.7 | 66.00 | 13.6 | 10.9 | 12.1 | 62.50 |
| | + CoMEM | **21.8** | **21.3** | **21.5** | 70.00 | **19.3** | **20.4** | **19.8** | **71.00** |
| Spanish | - | 17.3 | 16.5 | 16.9 | 75.79 | 16.7 | 16.7 | 16.7 | 72.64 |
| | + RAG | 12.3 | 11.4 | 11.8 | 79.25 | 9.5 | 8.1 | 8.7 | **76.10** |
| | + CoMEM | **24.0** | **23.3** | **23.6** | 79.87 | **23.0** | **21.8** | **24.3** | 75.47 |
| Portuguese | - | 18.7 | 18.1 | 18.4 | 66.55 | 18.4 | 19.3 | 18.8 | 66.90 |
| | + RAG | 15.8 | 13.8 | 14.7 | 62.32 | 13.7 | 13.5 | 13.6 | **70.07** |
| | + CoMEM | **27.1** | **27.2** | **27.2** | 66.90 | **24.1** | **26.1** | **25.1** | 67.96 |
| Bulgarian | - | 12.5 | 12.0 | 12.2 | 46.09 | 8.0 | 7.9 | 7.9 | 45.55 |
| | + RAG | 9.8 | 7.0 | 8.2 | 46.63 | 8.5 | 7.1 | 7.7 | 39.89 |
| | + CoMEM | **19.3** | **17.4** | **18.3** | **47.44** | **15.9** | **18.3** | **17.0** | **50.13** |
| Overall | - | 17.3 | 16.2 | 16.7 | 67.45 | 14.8 | 14.4 | 14.6 | 66.14 |
| | + RAG | 13.8 | 11.4 | 12.5 | 65.76 | 13.3 | 11.3 | 12.1 | 64.18 |
| | + CoMEM | **24.9** | **23.6** | **24.2** | **68.53** | **23.0** | **23.2** | **23.4** | **67.57** |

## 4.3 Further Analysis

**Long Context Understanding Study**  To evaluate the ability of models to handle long-context inputs, we compare our method against vanilla RAG under varying numbers of retrieved image-text

knowledge pairs. Specifically, we evaluate Qwen2-VL-Instruct and Qwen2.5-VL-Instruct on Infoseek, using both vanilla RAG and our method across different top-$k$ retrieval settings (from 3 to 50).

As shown in Figure 4, the results reveal a clear trend: RAG-based performance begins to degrade when more than 30 retrieved pairs are added, but our method remains stable and performs consistently well across all retrieval sizes. These findings show that discrete token-based methods struggle with long context, while continuous memory enables scalable and reliable long-context reasoning. This robust performance as context length increases underscores the advantage of our approach in processing long, information-dense inputs.

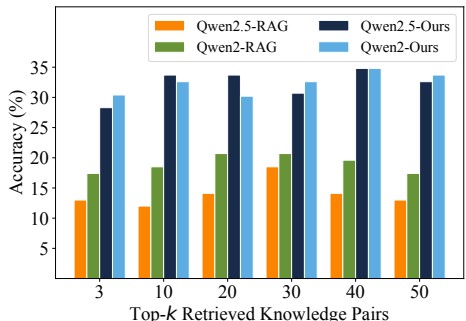

Figure 4: Comparison of Long Context Ability of RAG and Ours on Infoseek.

Table 5: Transferability Study of vision-language memory encoded by CoMEM on LLMs

| LLM | InfoSeek(%) | | | OVEN(%) | | | |
| --- | --- | --- | --- | --- | --- | --- | --- |
| | Unseen-Q | Unseen-E | All | Query | Entity | All | Avg. |
| Qwen2.5-Instruct | 5.0 | 4.8 | 4.9 | 2.4 | 0.1 | 1.3 | 3.1 |
| Qwen2.5-Instruct + RAG | 13.4 | 10.3 | 11.9 | 1.8 | 2.7 | 2.2 | 7.0 |
| Qwen2.5-Instruct + **CoMEM (using VLM)** | **29.3** | **27.4** | **28.3** | **6.8** | **7.7** | **7.2** | **17.8** |

**Transferability Study to LLMs.** To investigate whether the multimodal and multilingual continuous memory generated by a VLM can be effectively transferred to and leveraged by a pure Large Language Model (LLM), we conduct a transferability study. Specifically, we use Qwen2.5-VL-Instruct to encode visual and textual knowledge into dense continuous memory, and appended to the input embeddings of Qwen2.5-Instruct, a language-only LLM without vision capabilities.

We evaluate our approach on InfoSeek and OVEN. As shown in Table 5, our approach significantly outperforms both the vanilla LLM and the LLM augmented with text RAG, achieving an average accuracy of 17.8%, compared to 7.0% (RAG) and 3.1% (baseline). These results demonstrate that LLMs can effectively leverage VLM-generated memory, even without vision modules. This highlights a promising direction for cross-modal knowledge transfer, enabling LLMs to gain visual understanding through shared continuous memory without any architectural modifications.

# 5 Related Work

**Vision-Language Models.** LLMs have seen significant advancements, with models like GPT-4 [46] and Qwen-2.5 [52] demonstrating emergent capabilities such as in-context learning and complex reasoning. Building upon these advancements, VLMs have emerged to integrate visual and textual modalities, enabling models to process and understand multimodal data. To effectively extend language understanding into the visual domain, VLMs combine specialized neural network architectures for vision processing (such as Vision Transformers) with language models, enabling joint reasoning over visual and textual inputs. These models are typically trained on large-scale datasets that pair images with descriptive text to learn joint representations, using techniques like contrastive learning [53, 39], multimodal pretraining [54, 55], and instruction-aware tuning [47, 56, 57].

**Context Compression.** The constrained context windows of language models limit their information processing capacity, prompting the development of context compression methods to enable longer-sequence handling. One of the approach towards context compression in LLMs is through token pruning. FastV[14] distills vision-language knowledge into compact key-value memory slots, while SparseVLM[15] selects a sparse subset of visual tokens via top-down routing. In contrast, Gisting[58] compress long prompts into a small set of reusable "gist tokens" by modifying Transformer attention masks. Another approach involves soft prompts, which introduce trainable vector

embeddings to input sequences, enabling efficient task adaptation. IC-Former[59] compresses long input sequences into compact digest vectors, while SPC-LLM[60] combines natural language summarization with trainable soft prompts. Both methods condense lengthy input sequences into shorter representations, enhance the efficiency of LLMs and preserve over 90% of the original performance.

**Memory for Language Models.** As LMs face limitations in context length and long-term information retention, memory mechanisms have emerged to enhance their capacity for information-intensive reasoning and knowledge storage. Early retrieval-based approaches such as RAG [9] and REALM [11] retrieve external documents and inject them as long token sequences during inference time. However, these methods are constrained by context length limits and the inefficiency of discrete token representations, especially for supporting multimodal information. Recent advances shift toward continuous memory, representing knowledge as dense vectors rather than raw text. Approaches like VoCo-LLaMA [31] and MA-LMM [32] compress visual content into compact embeddings. Concurrently, strategies for memory storage have evolved. Persistent memory systems such as LONGMEM [10] store compressed knowledge in cache key-value (KV) formats, while retrieval-based methods like WikiLLaVA [38], RORA-VLM [28], and EchoSight [26] treat external knowledge bases as memory banks, using dedicated retrieval frameworks to support VQA tasks.

## 6    Conclusion

In this paper, we empirically demonstrate that a VLM can effectively serve as its own memory encoder, capable of converting multimodal knowledge into compact continuous embeddings. Building on this insight, we develop a data- and parameter-efficient method to fine-tune the VLM as a continuous memory encoder. Specifically, by updating only 1.2% of the model's parameters using just 15.6k self-synthesized samples, the resulting memory module can encode diverse multimodal and multilingual knowledge into merely 8 continuous embeddings. Importantly, since the VLM remains unchanged during inference, our memory module can be seamlessly integrated or detached as needed. Extensive evaluations across six English and two multilingual vision-reasoning benchmarks demonstrate the effectiveness and versatility of our approach.

In future work, we plan to extend our approach to a wider range of complex reasoning and planning tasks. Additionally, we aim to integrate the continuous memory mechanism into multimodal agents and evaluate its effectiveness in facilitating knowledge transfer across multiple language and vision-language models.

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

# A    Benchmark Details

**InfoSeek**    InfoSeek is a visual question answering (VQA) dataset tailored for information-seeking questions that cannot be answered with only common sense knowledge. It combines human-annotated and automatically collected data from visual entity recognition datasets and Wikidata, providing over one million examples for model fine-tuning and validation [35]. For InfoSeek, the ground truth answers for test sets are not publicly available, so we follow prior work [38, 26, 27] and report results on the validation sets. These sets include questions not seen during training and those associated with unseen entities.

**OVEN**    OVEN (Open-domain Visual Entity Recognition) challenges models to select among six million possible Wikipedia entities, making it a general visual recognition benchmark with the largest number of labels. It is constructed by re-purposing 14 existing datasets with all labels grounded onto one single label space: Wikipedia entities [42]. Similar with Infoseek, the ground truth answers for the test sets of OVEN are not publicly available, so we also report results on the validation sets.

**MRAG-Bench**    MRAG-Bench is a multimodal retrieval-augmented generation benchmark designed to evaluate the performance of large vision-language models (LVLMs) in scenarios where visual knowledge retrieval is more beneficial than textual information. It consists of 16,130 images and 1,353 human-annotated multiple-choice questions across nine distinct scenarios [43].

**OK-VQA**    OK-VQA includes more than 14,000 open-ended questions that require external knowledge to answer. The dataset is manually filtered to ensure all questions necessitate information beyond the image content, such as from Wikipedia [36].

**A-OKVQA**    A-OKVQA is a crowdsourced visual question answering dataset composed of approximately 25,000 questions requiring a broad base of commonsense and world knowledge to answer. Unlike existing knowledge-based VQA datasets, the questions generally cannot be answered by simply querying a knowledge base and instead require some form of commonsense reasoning about the scene depicted in the image [37].

**ViQuAE**    ViQuAE is a dataset focusing on knowledge-based visual question answering about named entities. It covers a wide range of entity types, such as persons, landmarks, and products, and evaluates models' abilities to ground visual content with knowledge base information [44].

**CVQA**    CVQA (Culturally-diverse Multilingual Visual Question Answering) dataset is a benchmark that offers a broad, inclusive representation by incorporating culturally-driven images and questions from a wide range of countries and languages[45]. In this study, we evaluate five of the most widely used languages in CVQA: Chinese, Russian, Spanish, Portuguese, and Bulgarian.

For all benchmarks, we follow the official evaluation protocols to compute the accuracy of the model's responses. Specifically: (1) For InfoSeek, OK-VQA, A-OKVQA, and ViQuAE, we use exact match evaluation to verify whether the model's response exactly matches the ground-truth answers. (2) For OVEN, we adopt the official evaluation script, which uses BM25 [61] to match the model's answer with relevant Wikipedia entities. (3) For MRAG-Bench and CVQA, which are in multiple-choice format, we evaluate accuracy by checking whether the model selects the correct option.

# B    Implementation Details

• **Knowledge Retrieval** Our knowledge base is constructed using the Wikipedia-based Image-Text (WIT) dataset[40], which consists of 37.5 million curated image-text pairs from Wikipedia articles across 108 languages. Based on WIT knowledge base, we implement a CLIP-based image-to-image retrieval system to identify the most relevant external knowledge. Following the stage-1 retrieval methodology of RoRA[28], we first encode all images in WIT using a frozen CLIP image encoder[39] to build a dense vector-search database. Given a query image $\mathcal{I}$, its CLIP embedding $CLIP(\mathcal{I})$ is compared against all vectors in the knowledge base via cosine similarity, followed by softmax normalization over the similarity scores. The image retriever then returns the top-$k$ highest-scoring images along with their associated textual descriptions.

• **Memory Encoding** Given the retrieved image-text pairs, we employ a memory encoder, consisting of a VLM and a Q-Former to compress multimodal information. For Each image-text pair is compressed into an 8-token vector. These token vectors are then concatenated and passed into the inference-time model. For Qwen2.5-Instruct VL, we uses Qwen2.5-Instruct VL as both the inference-time model and the memory encoder, and for Qwen2-Instruct VL, we uses Qwen2-Instruct VL as both the inference-time model and the memory encoder.

• **Answer Generation** The concatenated compressed tokens are plug into the inference-time model to generate answers. We should note that *our compression module is model-agnostic, allowing the memory encoder to be plugged into other LMs.* This flexibility is further demonstrated in Section 4.3.

## C  Ablation Study on Embedding Size

Table 6: Performance on Infoseek of Different Embedding Sizes

| Metric | 4-Emb | 8-Emb | 16-Emb | 24-Emb |
|---|---|---|---|---|
| Unseen Question Score | 29.32% | **32.80%** | 31.95% | 31.37% |
| Unseen Entity Score | 29.64% | **32.33%** | 30.03% | 30.83% |
| Final Score | 29.48% | **32.74%** | 30.96% | 31.10% |

To identify the optimal memory size for CoMEM, we conduct an ablation on the number of continuous embeddings (4, 8, 16, 24) using the InfoSeek [62] benchmark. As shown in Table 6, using 8 embeddings yields the best overall performance (32.74%), outperforming both smaller and larger configurations. Fewer embeddings underrepresent complex multimodal knowledge, while more embeddings introduce redundancy and noise. This result confirms that 8 embeddings offer an effective trade-off between expressiveness and efficiency, supporting our design choice in CoMEM.

## D  Method Generalizability

### D.1  Evaluation on Image Captioning

we evaluate the generalizability of CoMEM on a caption generation task using the COCO 2014 dataset [63]. We randomly sampled 100 image-caption pairs from the test set and used CLIP to retrieve relevant image-caption pairs from the training set. We then compared three setups: (1) Original model (no retrieval); (2) RAG-style retrieval (retrieved image-caption pairs prepended in prompt); (3) Our method (retrieved image-caption pairs encoded into continuous memory). We tested both Qwen2.5 and Qwen2, and report standard captioning metrics below in Table 7. We obtain the following conclusions:

**Better precision and fluency:**  BLEU-1 / BLEU-4 measure n-gram precision; METEOR balances precision and recall with synonym matching; higher scores indicate better word accuracy. CoMEM consistently improves BLEU-1, BLEU-4, and METEOR across both models, indicating more fluent and accurate captions.

**Substantial gain in content relevance:**  CIDEr scores use tf-idf weighted n-grams to measure relevance to ground-truth captions. It's increased significantly in Qwen2.5-VL with CoMEM (from $0.24 \rightarrow 0.64$), showing better content relevance to ground-truth captions.

**Improved semantic similarity:**  ROUGE-L captures the longest common subsequence with the reference, reflecting surface-level overlap. BERTScore-F computes semantic similarity using contextual embeddings. Memory-augmented models generate captions with high ROUGE-L and BERTScore-F scores, showing stronger lexical overlap and semantic similarity.

These results confirm that CoMEM generalizes well beyond QA tasks, offering an effective and lightweight memory mechanism for open-ended generation tasks like image captioning.

### D.2  Evaluation on Reasoning Tasks

Table 7: Evaluation Metrics for Qwen2.5-VL and Qwen2-VL on COCO Captioning task

| Model | Method | BLEU-1 | BLEU-4 | METEOR | CIDEr | ROUGE-L | BERTScore-F | Trigram Diversity |
|---|---|---|---|---|---|---|---|---|
| qwen2.5-VL | Original | 0.26 | 0.04 | 0.18 | 0.24 | 0.26 | 0.81 | 0.92 |
| | +RAG | 0.30 | 0.05 | 0.20 | 0.45 | 0.27 | 0.81 | **0.94** |
| | +CoMEM | **0.34** | **0.07** | **0.21** | **0.64** | **0.32** | 0.81 | 0.86 |
| qwen2-VL | Original | 0.34 | 0.08 | 0.21 | 0.79 | 0.34 | 0.81 | 0.84 |
| | +RAG | 0.34 | 0.07 | 0.20 | 0.64 | 0.32 | 0.81 | 0.83 |
| | +CoMEM | **0.36** | 0.08 | **0.23** | 0.76 | 0.32 | **0.82** | **0.88** |

Table 8: Performance Comparison on Reasoning Task MMMU

| Metrics | Method | Accounting | Architecture | Clinical Med | Computer | Economics | Electronics | Management | Materials | Pharmacy |
|---|---|---|---|---|---|---|---|---|---|---|
| Accuracy | Qwen2.5-VL | 26.7 | 23.3 | 43.3 | 26.7 | 46.7 | 23.3 | **40.0** | 30.0 | 30.0 |
| | +RAG | 23.3 | 26.7 | 40.0 | 33.3 | 33.3 | 16.7 | 36.7 | 33.3 | **40.0** |
| | +CoMEM | **40.0** | 26.7 | **43.3** | **40.0** | **46.7** | 30.0 | 36.7 | **33.3** | **40.0** |
| Similarity | Qwen2.5-VL | 26.7 | 26.7 | 43.3 | 26.7 | 46.7 | 26.7 | **40.0** | 30.0 | 30.0 |
| | +RAG | 23.3 | 26.7 | 40.0 | 33.3 | 33.3 | 16.7 | 36.7 | 33.3 | **40.0** |
| | +CoMEM | **40.0** | **33.3** | **43.3** | **40.0** | **50.0** | 30.0 | **43.3** | **33.3** | **43.3** |

In recent LLM/MLLM reasoning studies, various reasoning-heavy benchmarks such as MathVista [64] and MMMU [65] have been widely adopted. It would be interesting to see whether the proposed method can effectively handle reasoning tasks on these benchmarks.

Concretely, we split the dataset into evaluation set and retrieval set, and retrieve relevant image-text pairs using CLIP in RAG and Memory settings. We calculate accuracy with exact matching and similarity with overlapping ratio between predicted and ground truth tokens.

Table 9: Performance Comparison on Reasoning Task MathVista

| Metrics | Method | Math | Textbook QA | Visual QA | Figure QA |
|---|---|---|---|---|---|
| Accuracy | Qwen2.5-VL | 59 | 50 | 34 | 46 |
| | +RAG | 63 | **53** | 37 | 54 |
| | +CoMEM | **65** | 46 | **39** | **56** |
| Similarity | Qwen2.5-VL | 62 | 55 | 35 | 48 |
| | +RAG | 63 | **57** | 37 | 56 |
| | +CoMEM | **66** | 49 | **41** | **57** |

Our results show that CoMEM consistently improves performance across diverse domains:

On MMMU (Table 8), our method consistently outperforms both the Original and RAG baselines across most domains, especially in domains like Accounting, Computer Science, Pharmacy, Electronics, etc. On MathVista (Table 9), we also observe improvements in different types of reasoning-heavy tasks. These results demonstrate that our proposed method effectively enhances accuracy in complex reasoning scenarios across domains, showing the robustness and generalizability of our method.

# E  Method Cost

## E.1  Inference Latency Test

To test the latency of baselines and our method, we evaluated the throughput (tokens per second) for each model on 100 random sampled instances. The higher throughput indicates faster inference speed. All experiments were conducted on a single NVIDIA H100 and the results are summarized in Table 10.

The findings show that CoMEM maintains competitive inference speed as the original model, because it does not significantly increase input length. For RAG, as it expands the input by up to 15× and causes a proportional increase in attention cost, the inference speed decreases a lot. Since our CoMEM adds less than 100 continuous memory tokens, this keeps inference cost close to that of the base model while improving accuracy.

## E.2  Token Cost Comparison

Continuous embeddings better support high compression, as continuous embeddings can densely encode the information, which prevents the great increase of context length (see Table11). It is rather helpful for VLMs to read and understand massive multimodal data. For the example of RAG scenario,

Table 10: Inference Latency Evaluation accross different benchmarks

| Token per Second | InfoSeek | OVEN | MRAG | OKVQA | AOKVQA | ViQuAE |
|---|---|---|---|---|---|---|
| **Qwen2 models** | | | | | | |
| Qwen2-VL | 55.93 | 100.91 | 39.54 | 45.81 | 53.38 | 99.53 |
| Qwen2-VL+RAG | 40.71 | 49.27 | 8.38 | 34.66 | 23.60 | 37.50 |
| Qwen2-VL+COMEM | 48.93 | 53.29 | 18.61 | 40.42 | 51.60 | 41.50 |
| **Qwen2.5 models** | | | | | | |
| Qwen2.5-VL | 53.44 | 72.71 | 42.13 | 45.64 | 42.87 | 53.49 |
| Qwen2.5-VL+RAG | 41.17 | 42.04 | 15.89 | 32.22 | 27.93 | 39.61 |
| Qwen2.5-VL+CoMEM | 51.05 | 54.39 | 19.79 | 42.86 | 61.41 | 48.81 |

Table 11: Token Number and Accuracy Across Datasets

| Token Number (Accuracy%) | Infoseek | Oven | MRAG | OKVQA | AOKVQA | ViQuAE |
|---|---|---|---|---|---|---|
| **Qwen2-VL** | 357.6 (17.8) | 408.6 (25.5) | 84.5 (39.3) | 404.8 (36.3) | 418.4 (41.8) | 424.7 (34.5) |
| **Qwen2-VL+RAG** | 6163.8 (19.0) | 6481.5 (24.7) | 5882.4 (40.4) | 6803.8 (41.9) | 6715.4 (45.3) | 6943.2 (33.6) |
| **Qwen2-VL+COMEM** | 416.8 (33.1) | 492.9 (30.5) | 176.9 (35.1) | 453.8 (57.7) | 478.3 (60.6) | 489.2 (36.3) |

here we list the context length in three settings: Original model(no retrieval) , Qwen2-VL+RAG and our CoMEM method. We can clearly see that although we only retrieve 10 examples in the context, the context length increases by approximately 15× when using RAG. It will cause the VLM to suffer from the long context understanding problem. In contrast, our method converts the retrieved examples into continuous embeddings, which leads to fewer than 100 additional embeddings to the input sequence. It avoids the long context understanding problem and also reduces the cost of encoding such long input sequences.

# F  Training Efficiency

To evaluate the training efficiency of our method, we assess the performance of CoMEM on Qwen2.5-VL using the Infoseek benchmark under varying amounts of training data and trainable parameters. In the original setting, we use only 15.6k training samples and fine-tune 1.2% of the total parameters. For the data variation setting, we scale the training data by factors of 0.25×, 0.5×, 2×, and 4×. For the parameter variation setting, we adjust the LoRA rank and the number of Q-Former layers by the same scaling factors to control the number of trainable parameters.

| Training Settings | | Infoseek | | |
|---|---|---|---|---|
| | | Unseen-Q | Unseen-E | All |
| Original | | 32.8 | 28.5 | 30.7 |
| Data | 4x | **34.8** | 28.4 | **31.3** |
| | 2x | 32.2 | **29.8** | 30.9 |
| | 0.5x | 26.5 | 24.4 | 25.4 |
| | 0.25x | 17.8 | 17.5 | 17.6 |
| Parameters | 4x | 26.4 | 22.1 | 24.1 |
| | 2x | 28.6 | 24.8 | 26.3 |
| | 0.5x | 27.8 | 24.7 | 26.1 |
| | 0.25x | 23.1 | 20.3 | 21.6 |

Table 12: Performance of CoMEM on Qwen2.5-VL under different training data and parameter settings.

As shown in Table 12, increasing the training data by 2× or even 4× results in only marginal performance gains, suggesting that the original data size is already adequate for effective training. Similarly, increasing the number of trainable parameters does not yield improvements, while reducing them below the original configuration leads to a notable drop in performance. These findings highlight that our training recipe is both data- and parameter-efficient, achieving strong results with minimal resource expenditure.

## G  Limitations

• **Evaluation Benchmarks** While we evaluate our method on 6 multimodal and 2 multilingual reasoning tasks, most of benchmarks are static and synthetic. Real-world applications with dynamic or noisy inputs (*e.g.,* web data, live video) may introduce challenges.

• **Multi-Agent Settings** Our current framework is designed and evaluated in a single-model setting, where one inference language model uses the continuous memory module for enhanced reasoning. However, many real-world applications involve multiple collaborating agents or a combination of LMs and VLMs. Whether our continuous memory can effectively transmit and share knowledge across multiple models remains unexplored and will be investigated in future work.

## H  Case Study

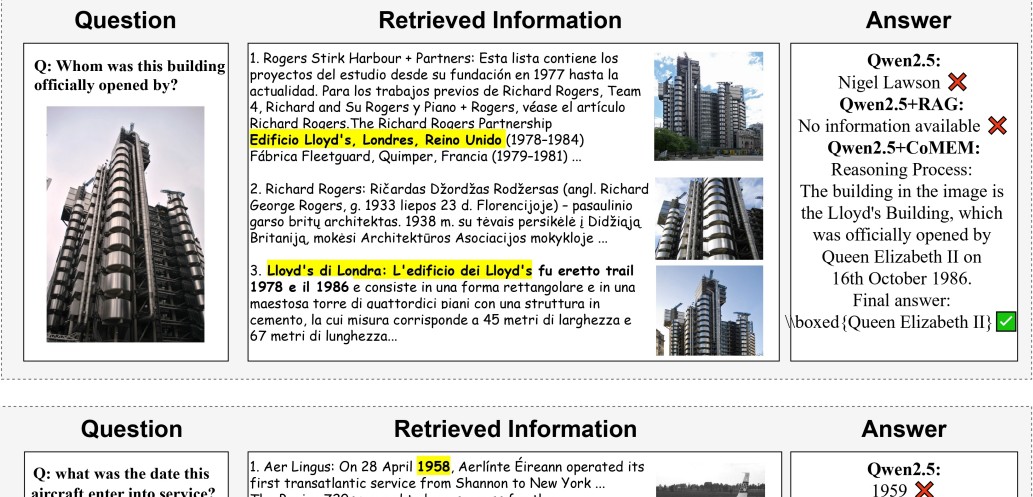

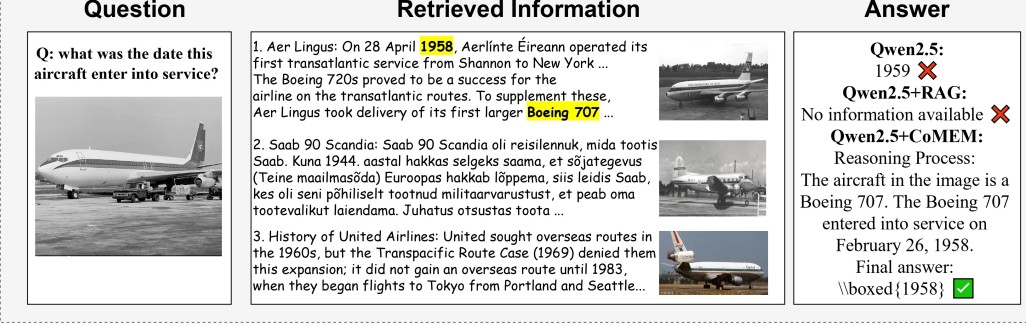

Figure 5: Case studies comparing CoMEM with baseline model and model with RAG.

In this appendix, we present a qualitative case study to demonstrate the effectiveness of our proposed model. Given a question and a corresponding query image, our pipeline first retrieves the top 10 relevant image-text pairs from the WIT knowledge base to provide rich contextual information. Due to space constraints, we only display three representative retrieved pairs for each example in this appendix. We then compare the performance of our CoMEM model against two baselines: the standalone Qwen2.5-VL and a baseline retrieval-augmented generation (RAG) model. CoMEM can effectively capture key information from retrieved supporting texts, even when the exact answer is not explicitly provided, and perform reasoning to derive the correct answer.

These case studies demonstrate that CoMEM is able to generate accurate answers in challenging scenarios where baseline models either fail or return incomplete information. This highlights CoMEM's ability to effectively encode and leverage complex multimodal and multilingual knowledge, leading to stronger performance in advanced reasoning tasks.

