# OpenReview forum: "Towards General Continuous Memory for Vision-Language Models"
_NeurIPS.cc/2025/Conference — NeurIPS 2025 poster_

### Official Review · Reviewer_u2aH · 2025-06-07

**Clarity:** 4
**Significance:** 3
**Originality:** 3
**Rating:** 5
**Confidence:** 3

**Summary:**

This paper proposes an innovative continuous memory mechanism to enhance the performance of Vision-Language Models (VLMs) in complex multimodal and cross-lingual reasoning tasks. While RAG concatenates multimodal information as long-sequence inputs, it often leads to excessive context length and performance degradation.
This work is the first to leverage VLMs themselves as memory encoders, eliminating the need for costly semantic alignment training.
Experimental results demonstrate an average accuracy improvement of 5.1%–8.0% across eight multimodal reasoning benchmarks (e.g., OK-VQA, A-OKVQA), with stable performance in long-context scenarios.

**Questions:**

See weakness. I am open to discuss. If the author can clarify my questions, I am happy to improve my score.

**Ethical Concerns:**

["NO or VERY MINOR ethics concerns only"]

**Final Justification:**

I appreciate the authors' clarification, which has addressed my concerns.

**Limitations:**

There are no potential moral and ethical issues.

**Paper Formatting Concerns:**

No.

**Quality:**

3

**Strengths And Weaknesses:**

**Strength:**
1. Proposes VLM-based auto-encoding continuous memory.
2. Introduces CoMEM, an efficient training framework:

    (a) Parameter-efficient: Tunes only 1.2% of parameters (via LoRA and lightweight Q-Former).

    (b) Data-efficient: Requires only 15.6K self-generated samples (synthesized by VLM).

    (c) Modular design: Keeps VLM frozen during inference, with plug-and-play memory modules.

3. Extensive experiments demonstrate effectiveness, particularly excelling in low-resource languages (e.g., Bulgarian, Russian).

**Weakness:**
1. Is the method proposed in this paper still effective for models other than Qwen2?

2. The paper's motivation stems from the concern that RAG "may drastically increase context length and even degrade performance." However, Table 3 shows non-robust performance changes after applying RAG. Perhaps highlighting which data are long-context in Table 3 could better align the results with the motivation.

---

> ### Author Rebuttal · Authors · 2025-07-30
>
> ## Rebuttal
>
> We sincerely appreciate the reviewer’s thoughtful feedback and are grateful for the encouraging remarks. Below, we reply to all the concerns mentioned in the weaknesses and questions section.
>
> ### **Weakness-1: Unclear Model Generalization**
> > The effectiveness of the proposed method is primarily demonstrated on Qwen2, leaving it unclear how well the approach generalizes to other models.
>
> **Reply-to-Weakness1**:
>
> Our method is effective for equipping models other than Qwen2-VL to improve the performance, even helpful for augmenting Qwen2-LLM with multimodal knowledge.
> - **Improved Performance on Qwen2-VL and Qwen2.5-VL**: In our paper we evaluate effectiveness of CoMEM on Qwen2-VL and Qwen2.5-VL (Tables 3, 4 in paper). Here we add result table1 by comparing with up-to-date RAG methods. The result shows that CoMEM significantly improves performance over both the original models and strong RAG baselines, demonstrating consistent gains across different models.
> - **Multimodal Knowledge Transfer to Text-only LLM**: In Section 4 (Transferability Study to LLMs), we demonstrate that a text-only LLM (Qwen2.5-Instruct) can leverage the multimodal knowledge provided by our CoMEM using Qwen2.5-VL as the encoder. The result is in Table2. It shows that CoMEM is a powerful and general bridge between VLMs and LLMs, empowering the visual reasoning ability of LLMs, and validating the generalizability of our approach.
>
> **Table1: Performance Comparison Across Datasets (Accuracy %)**
>
> | Model              | infoseek(Q) | infoseek(E) | OVEN    | MRAG    | OKVQA    | AOKVQA     | ViQUAE   |
> |--------------------|-------------|-------------|---------|---------|----------|------------|----------|
> | Wiki-LLaVA         | 28.6        | 25.7        | –       | –       | –        | –          | –        |
> | RORA-VLM           | 27.3        | 25.1        | 15.5    | –       | –        | –          | –        |
> | REVEAL             | –           | –           | –       | –       | **59.1** | *52.2*     | –        |
> | Echosight          | 18.0        | 19.8        | **27.0**| **41.3**| 20.0     | 16.9       | 25.2     |
> | ReflectiVA         | 28.6        | *28.1*      | 20.6    | *39.7*  | *47.5*   | 47.6       | *29.8*   |
> | Qwen2+CoMEM        | *32.6*      | **33.1**    | *23.6*  | 35.1    | *57.7*   | **60.6**   | **36.3** |
> | Qwen2.5+CoMEM      | **32.8**    | 28.5        | 20.8    | 38.1    | 47.6     | *55.0*     | 34.7     |
>
> **Table2: Transferability Study of Vision-Language Memory Encoded by CoMEM on LLMs (Accuracy %)**
>
> | LLM                         | InfoSeek: Query | InfoSeek: Entity | InfoSeek: All | OVEN: Query | OVEN: Entity | OVEN: All | Avg. |
> |----------------------------|--------------------|---------------------|----------------|--------------|---------------|------------|-------|
> | Qwen2.5-Instruct           | 5.0                | 4.8                 | 4.9            | 2.4          | 0.1           | 1.3        | 3.1   |
> | Qwen2.5-Instruct + RAG     | 13.4               | 10.3                | 11.9           | 1.8          | 2.7           | 2.2        | 7.0   |
> | Qwen2.5-Instruct + CoMEM   | **29.3**           | **27.4**            | **28.3**       | **6.8**      | **7.7**       | **7.2**    | **17.8** |
>
> ### **Weakness-2: Motivation is not shown in Result clearly**
> > The paper motivates the method by stating that RAG may increase context length and degrade performance, but Table 3 shows only modest or inconsistent performance changes with RAG. Highlighting which samples involve long contexts in Table 3 could help better connect the empirical results to the stated motivation.
>
> **Reply-to-Weakness2**:
>
> We sincerely thank the reviewer for this insightful suggestion and will report the average context length for each benchmark with or without our method. These clearly show that RAG makes all the inputs of these benchmarks become long-context, approximately **15×** times (see Table 3 below). As long context understanding is a hard problem for LLMs, the performance is also affected:
> - **RAG performance is unstable due to the long context**: While RAG occasionally improves performance (e.g., Qwen2-VL on OKVQA: 36.3 → 41.9), it fails to deliver consistent gains across benchmarks, and even degrades the performance (e.g., Qwen2.5-VL on AOKVQA: 39.8 → 34.9). It indicates that although the additional long context contains useful reference data, the lengthy context makes it difficult for VLM to read and use this information.
> - **CoMEM performs better across different benchmarks, benefiting from its compression ability**: CoMEM provides consistent gains on most of the benchmarks, while using much fewer tokens. This reinforces our claim that our unified memory is not only more efficient but also reliable across diverse tasks.
>
> **Table3: Token Number and Accuracy Across Datasets**
>
> | **Token Number (Accuracy %)** | **Infoseek**       | **OVEN**          | **MRAG**          | **OKVQA**         | **AOKVQA**        | **ViQuAE**        |
> |-----------------------------|--------------------|-------------------|-------------------|-------------------|-------------------|-------------------|
> | **Qwen2-VL**                | 357.6 (17.8)      | 408.6 (25.5)     | 84.5 (39.3)      | 404.8 (36.3)      | 418.4 (41.8)      | 424.7 (34.5)      |
> | **Qwen2-VL+RAG**            | 6163.8 (19.0)     | 6481.5 (24.7)    | 5882.4 (40.4)    | 6803.8 (41.9)     | 6715.4 (45.3)     | 6943.2 (33.6)     |
> | **Qwen2-VL+COMEM**          | 416.8 (**33.1**)       | 492.9 (**30.5**)      | 176.9 (**35.1**)      | 453.8 (**57.7**)      | 478.3 (**60.6**)      | 489.2 (**36.3**)      |

---

> > ### Comment · Reviewer_u2aH · 2025-08-03
> > **Acknowledgement of rebuttal from reviewer u2aH**
> >
> > I appreciate the authors' clarification, which has addressed my concerns. However, I maintain that Qwen 2.5 and Qwen 2 belong to the same category of models, and I encourage the authors to include results from additional models in the next version of the paper. Nevertheless, I find this to be a good paper and commend the authors for their efforts. I am willing to raise my score from 4 to 5. Best of luck!

---

### Official Review · Reviewer_W84i · 2025-07-01

**Clarity:** 3
**Significance:** 2
**Originality:** 3
**Rating:** 4
**Confidence:** 3

**Summary:**

This work aims to enhance the capability of vision-language models (VLMs) in acquiring multimodal or multilingual real-world knowledge. To efficiently leverage an external memory system, the paper proposes using continuous memory, i.e., a compact set of dense embeddings, to represent multimodal or multilingual knowledge. Specifically, the authors show that a VLM can serve as its own continuous memory encoder. Based on this observation, the paper introduces a data-efficient and parameter-efficient method to fine-tune the VLM into a memory encoder. Extensive experiments show that the proposed method achieves strong performance with minimal additional parameters.

**Questions:**

Please see the weakness.

**Ethical Concerns:**

["NO or VERY MINOR ethics concerns only"]

**Final Justification:**

Thanks for the authors' detailed response. My concerns have been addressed, so I have increased the score to 4.

**Limitations:**

yes

**Quality:**

2

**Strengths And Weaknesses:**

**Strength**

The paper provides an empirical study demonstrating that a VLM can serve as a continuous memory encoder to compress multimodal information, and that a few embeddings from the VLM can preserve key information to improve performance on multimodal reasoning tasks.

**Weakness**

I am quite confused about the memory mechanism introduced in this work. From a high-level technical perspective, the proposed framework seems to be a LoRA-based fine-tuning framework with retrieved images/texts. A typical memory system should involve key processes such as information storage, updating, and retrieval. These aspects are not clearly illustrated in the paper, and the current presentation makes the proposed method appear more like a LoRA-based fine-tuning approach rather than a complete memory mechanism.

In the inference stage, are only the memory embeddings used for making predictions, while the rest of the VLM’s parameters are unused? The claim that “we keep the VLM’s parameters frozen during inference, while continuous memory embeddings are directly used for downstream tasks” is unclear. The authors should provide more details about the inference stage and clarify how the memory contributes to prediction.

The compared methods are somewhat limited. In the SOTA table, only vanilla RAG is compared. However, various multimodal RAG methods have been studied [a]. The authors should conduct a more comprehensive comparison with other RAG-based approaches to better validate the effectiveness of their method.

[a] Ask in Any Modality: A Comprehensive Survey on Multimodal Retrieval-Augmented Generation.

In recent LLM/MLLM reasoning studies, various reasoning-heavy benchmarks such as MathVista and MMMU have been widely adopted. It would be interesting to see whether the proposed method can effectively handle reasoning tasks on these benchmarks.

Minor: The authors could consider moving Figure 1 and Table 1 closer to the sections where the relevant concepts are first discussed (e.g., the Introduction), to improve readability.

---

> ### Author Rebuttal · Authors · 2025-07-31
>
> ## **Rebuttal - Part 1**
>
> We thank the reviewer for the detailed and constructive feedback. We respectfully reply to all the concerns raised in the weaknesses and questions section below.
>
> ### **Question-1: Unclear Memory Mechanism**
> > The proposed method is presented as a memory system, but key aspects typically associated with memory—such as information storage, updating, and retrieval—are not clearly defined or illustrated. From a high-level perspective, the method appears more like a LoRA-based fine-tuning framework with retrieved image/text inputs, rather than a complete or standalone memory architecture.
>
> **Reply-to-Question1**:
>
> We appreciate the reviewer’s thoughtful question and would like to clarify our design and contributions as follows:
>   - **Memory Encoder, Not a Full Memory System**: Our work proposes a memory encoder module, rather than a complete memory system. Its primary goal is to encode external knowledge into compact continuous embeddings that can be flexibly integrated into the VLM to enhance its capabilities.
>
>   - **Plug-and-play and Generalizable Memory**: As demonstrated in Table 3 in the paper, the proposed memory encoder is effective across different backbones, such as Qwen2-VL and Qwen2.5-VL. Furthermore, Table 1 below shows that our continuous memory can successfully transfer visual and textual knowledge to text-only LLMs. This indicates that the continuous memory representations are broadly compatible and can effectively preserve transferable multimodal knowledge for augmenting any LLM.
>
>   - **Simple Yet Effective Retrieval**: While our current implementation adopts a CLIP-based retrieval mechanism, consistent with prior work, our method already demonstrates strong performance. We believe that incorporating more advanced retrieval strategies could further enhance the results.
>
>   - **Storage Efficiency**: Our approach offers a significant storage advantage over traditional designs. The following Table 2 shows the average context length for each benchmark with or without our method. It clearly shows that the discrete token format is approximately 15x times larger than the continuous memory format. By storing compact continuous embeddings instead of original images or discrete token sequences, we can substantially reduce memory and storage requirements.
>
> **Table 1: Transferability Study of Vision-Language Memory Encoded by CoMEM on LLMs**
>
> | LLM                        | InfoSeek: Unseen-Q | InfoSeek: Unseen-E | InfoSeek: All | OVEN: Query | OVEN: Entity | OVEN: All | Avg. |
> |---------------------------|--------------------|---------------------|----------------|--------------|---------------|-------------|--------------|
> | Qwen2.5-Instruct          | 5.0                | 4.8                 | 4.9            | 2.4          | 0.1           | 1.3         | 3.1          |
> | Qwen2.5-Instruct + RAG    | 13.4               | 10.3                | 11.9           | 1.8          | 2.7           | 2.2         | 7.0          |
> | Qwen2.5-Instruct + CoMEM  | **29.3**           | **27.4**            | **28.3**       | **6.8**      | **7.7**       | **7.2**     | **17.8**     |
>
>
> **Table 2: Token Number and Accuracy Across Datasets**
>
> | **Token Number (Accuracy)** | **Infoseek**       | **Oven**          | **MRAG**          | **OKVQA**         | **AOKVQA**        | **ViQuAE**        |
> |-----------------------------|--------------------|-------------------|-------------------|-------------------|-------------------|-------------------|
> | Qwen2-VL                | 357.6 (17.8)      | 408.6 (25.5)     | 84.5 (39.3)      | 404.8 (36.3)      | 418.4 (41.8)      | 424.7 (34.5)      |
> | +Multimodal Knowledge in Discrete Tokens            | 6163.8 (19.0)     | 6481.5 (24.7)    | 5882.4 (**40.4**)    | 6803.8 (41.9)     | 6715.4 (45.3)     | 6943.2 (33.6)     |
> | +Multimodal Knowledge in Continuous Memory          | 416.8 (**33.1**)       | 492.9 (**30.5**)      | 176.9 (35.1)      | 453.8 (**57.7**)      | 478.3 (**60.6**)      | 489.2 (**36.3**)      |
>
> ### **Question-2: Inference Process Ambiguity**
> > It is unclear how the memory embeddings interact with the VLM during inference. The claim that “we keep the VLM’s parameters frozen during inference, while continuous memory embeddings are directly used for downstream tasks” lacks clarity. More detailed explanation of the inference workflow and the contribution of memory in prediction is needed.
>
> **Reply-to-Question2**:
>
> We apologize for the confusion; there was a typo in the original statement. The correct description is:  “We keep the VLM’s parameters frozen during training, and only train the LoRA adapters and the Q‑Former.” Below we clarify both the training and inference processes:
>   - **Training stage**:
>     - **Trainable Components**: (i) LoRA adapters (low‑rank update matrices) in selected layers of the VLM encoder and (ii) the Q‑Former, which compresses multimodal knowledge items into 8 continuous memory embeddings each.
>     - **Frozen Components**: All original VLM parameters are frozen.
>
>     This design updates only 1.2% of parameters, following a parameter‑efficient fine‑tuning strategy (Section 3.3, “Parameter-efficient Fine-tuning”).
>   - **Inference stage**:
>     - The trained Q‑Former generates continuous memory embeddings for retrieved multimodal knowledge items. These memory embeddings are then prepended to the input embedding sequence of the frozen VLM(Section 3.2, “Plug‑and‑Play Mechanism”). The VLM then performs normal autoregressive generation, attending to the memory embeddings and the query jointly.
>     - Therefore, the continuous memory is consumed by the VLM’s self‑attention layers during decoding, which provide retrieved knowledge to guide prediction. This design allows the memory module to be plug‑and‑play without altering the base VLM’s parameters. As shown in Table 2 in the paper, this paradigm achieves the lowest training cost among most state‑of‑the‑art RAG models while still delivering superior performance.
>
> ### **Question-3: Limited Comparison Baselines**
> > In the SOTA table, only vanilla RAG is compared. The authors should conduct a more comprehensive comparison with other RAG-based approaches to better validate the effectiveness of their method.
>
> **Reply-to-Question3**:
>
> Thank you for your suggestion. We agree that stronger and more diverse baselines are important for a fair evaluation, and we have selected SOTA ones in our paper for comparison.
>   - **Competitive RAG methods**: In Table 3 (in paper), we have compared four highly relevant and latest multimodal RAG-based methods: Wiki-LLaVA, RORA-VLM, EchoSight, and ReflectiVA.
>   - **Adding More Experiments on RAG Baselines**: To address your concern, we have: (1) Re-implemented EchoSight and ReflectiVA (the only open-sourced methods among the four) and tested it on all benchmarks (MRAG, OKVQA, AOKVQA, ViQUAE) to provide a comprehensive evaluation. (2) Included more relevant RAG-based methods for comparison:
>     - **Wiki-LLaVA**[1]: Uses hierarchical retrieval from Wikipedia to add external knowledge for visual question answering.
>     - **RORA-VLM**[2]: Enhances robustness with two-stage retrieval and adversarial training to filter irrelevant content.
>     - **EchoSight**[3]: Applies a visual-first retrieval followed by text-image reranking for better encyclopedic knowledge integration.
>     - **ReflectiVA**[4]: Uses reflective tokens to decide when and what external knowledge to retrieve.
>     - **REVEAL**[5]: Pre-trains a retrieval-augmented VLM with a large multimodal memory for knowledge-intensive tasks.
>
>     The results are shown in the following Table 3. This extended evaluation demonstrates that CoMEM consistently outperforms open-source multimodal RAG methods (Echosight, ReflectiVA) across tasks, and is competitive with or superior to closed-source systems like RORA-VLM, REVEAL, etc. These results further validate the generality, scalability, and effectiveness of our proposed memory-based method.
>
>
> **Table 3: Performance Comparison Across Datasets**
>
> | Model              | infoseek(Q) | infoseek(E) | OVEN  | MRAG  | OKVQA | AOKVQA | ViQUAE |
> |--------------------|-------------|-------------|--------|--------|--------|---------|--------|
> | Wiki-LLaVA         | 28.6        | 25.7        | –      | –      | –      | –       | –      |
> | RORA-VLM           | 27.3        | 25.1        | 15.5   | –      | –      | –       | –      |
> | REVEAL             | –           | –           | –      | –      | 59.1   | 52.2    | –      |
> | Echosight          | 18.0        | 19.8        | 27.0   | 41.3   | 20.0   | 16.9    | 25.2   |
> | ReflectiVA         | 28.6        | 28.1        | 20.6   | 39.7   | 47.5   | 47.6    | 29.8   |
> | Qwen2-VL+CoMEM        | 32.6        | 33.1        | 23.6   | 35.1   | 57.7   | 60.6    | 36.3   |
> | Qwen2.5-VL+CoMEM      | 32.8        | 28.5        | 20.8   | 38.1   | 47.6   | 55.0    | 34.7   |
>
> [1] Davide Caffagni et al. Wiki-LLaVA: Hierarchical Retrieval-Augmented Generation for Multimodal LLMs. In CVPR 2024.
>
> [2] Jingyuan Qi et al. RoRA-VLM: Robust Retrieval-Augmented Vision Language Models. In 2024.
>
> [3] Yibin Yan . EchoSight: Advancing Visual-Language Models with Wiki Knowledge. In EMNLP 2024.
>
> [4] Federico Cocchi et al. Augmenting Multimodal LLMs with Self-Reflective Tokens for Knowledge-based Visual Question Answering. In CVPR 2025.
>
> [5] Ziniu Hu et al. REVEAL: Retrieval-Augmented Visual-Language Pre-Training with Multi-Source Multimodal Knowledge Memory. In CVPR 2023.

---

> > ### Comment · Reviewer_W84i · 2025-08-05
> >
> > Thank the authors for the response. Most of my concerns have been addressed. I will take the response into account when adjusting my score.

---

> > > ### Author Response · Authors · 2025-08-05
> > >
> > > We are sincerely grateful for the reviewer’s feedback! Thank you for considering our clarifications in your response. If there are any additional questions you'd like us to further elaborate on, we would be more than happy to provide additional details and continue the discussion. We greatly value your feedback and hope that the improvements can be reflected in your final assessment.

---

> > > ### Author Response · Authors · 2025-08-06
> > >
> > > Dear Reviewer,
> > >
> > > We sincerely appreciate your feedback and aim to fully address your concerns.
> > >
> > > During the rebuttal phase, we split our response into two parts—one in the main rebuttal and another in the Author AC Confidential Comment section, which we now realize is not visible to reviewers. To provide complete context, we’re including the previously hidden portion below.
> > >
> > > We do hope we can address your concerns and would be happy to engage in further discussion if you have any follow-up questions. We truly appreciate your feedback and hope our improvements are reflected in your final evaluation.
> > >
> > > ## Rebuttal - Part 2
> > >
> > > ### **Question-4: Generalization on Reasoning Tasks**
> > > > In recent LLM/MLLM reasoning studies, various reasoning-heavy benchmarks such as MathVista and MMMU have been widely adopted. It would be interesting to see whether the proposed method can effectively handle reasoning tasks on these benchmarks.
> > >
> > > **Reply-to-Question4**:
> > >
> > > Following the suggestion of the reviewer, we evaluate our method on reasoning-heavy benchmarks like MathVista and MMMU. Concretely, we split the dataset into evaluation set and retrieval set, and retrieve relevant image-text pairs using CLIP in RAG and Memory settings. We calculate accuracy with exact matching and similarity with overlapping ratio between predicted and ground truth tokens.
> > >
> > >   Our results show that CoMEM consistently improves performance across diverse domains:
> > >   - **On MMMU** (Table 4), our method consistently outperforms both the Original and RAG baselines across most domains, especially in domains like Accounting, Computer Science, Pharmacy, Electronics, etc.
> > >   - **On MathVista** (Table 5), we also observe improvements in different types of reasoning-heavy tasks. These results demonstrate that our proposed method effectively enhances accuracy in complex reasoning scenarios across domains, showing the robustness and generalizability of our method.
> > >
> > > **Table 4: Performance Comparison on Reasoning Task MMMU**
> > >
> > > | Metrics    | Method   | Accounting | Architecture | Clinical Med | Computer | Economics | Electronics | Management | Materials | Pharmacy |
> > > |------------|----------|------------|---------------|---------------|-------------|------------|--------------|-------------|------------|-----------|
> > > | Accuracy   | Qwen2.5-VL | 26.67      | 23.33         | 43.33         | 26.67       | 46.67      | 23.33        | **40.00**      | 30.00         | 30.00        |
> > > |            | +RAG      | 23.33      | 26.67         | 40.00         | 33.33       | 33.33      | 16.67        | 36.67       | 33.33      | **40.00**     |
> > > |            | +CoMEM   | **40.00**     | 26.67         | **43.33**     | **40.00**      | **46.67**  | **30.00**       | 36.67       | **33.33**  | **40.00**     |
> > > | Similarity | Qwen2.5-VL  | 26.67      | 26.67         | 43.33         | 26.67       | 46.67      | 26.67        | **40.00**      | 30.00         | 30.00        |
> > > |            | +RAG      | 23.33      | 26.67         | 40.00         | 33.33       | 33.33      | 16.67        | 36.67       | 33.33      | **40.00**     |
> > > |            | +CoMEM   | **40.00**     | **33.33**     | **43.33**     | **40.00**      | **50.00**     | **30.00**       | **43.33**   | **33.33**  | **43.33**  |
> > >
> > > **Table 5: Performance Comparison on Reasoning Task MathVista**
> > >
> > > | Metrics    | Method   | Math | Textbook QA | Visual QA | Figure QA |
> > > |------------|----------|------|--------------|------------|------------|
> > > | Accuracy   | Qwen2.5-VL | 59   | 50           | 34         | 46         |
> > > |            | +RAG      | 63   | **53**       | 37         | 54         |
> > > |            | +CoMEM   | **65** | 46         | **39**     | **56**     |
> > > | Similarity | Qwen2.5-VL | 62   | 55        | 35         | 48      |
> > > |            | +RAG      | 63   | **57**    | 37         | 56      |
> > > |            | +CoMEM   | **66** | 49      | **41**     | **57**  |

---

> > > > ### Comment · Reviewer_W84i · 2025-08-07
> > > >
> > > > Thanks for considering my additional suggestions. The results look good, and I will raise my score accordingly.

---

> > > > > ### Author Response · Authors · 2025-08-08
> > > > >
> > > > > Dear Reviewer,
> > > > >
> > > > > Thank you very much for considering our responses and for your kind words! We truly appreciate your thoughtful feedback and your intention to raise the score—it means a lot to us.
> > > > >
> > > > > We just want to gently follow up in case the score update was overlooked. Your support is very encouraging and valuable to us, and we’re grateful for the time and effort you’ve dedicated to reviewing our work.
> > > > >
> > > > > Thank you again for your support!
> > > > >
> > > > > Best regards,
> > > > >
> > > > > The authors of Paper *Towards General Continuous Memory for Vision-Language Models*

---

> > > > > > ### Author Response · Authors · 2025-08-09
> > > > > > **Thank you**
> > > > > >
> > > > > > Dear Reviewer W84i,
> > > > > >
> > > > > > As the rebuttal period is coming to a close, we would like to sincerely thank you for your valuable feedback and thoughtful suggestions. We truly appreciate the time and effort you’ve taken to engage with our work—your input has been incredibly helpful in guiding us to improve the paper.
> > > > > >
> > > > > > Thank you again for your support!
> > > > > >
> > > > > > Sincerely,
> > > > > > The Authors

---

> ### Author Response · Authors · 2025-08-07
>
> Thank you very much for considering our responses! We truly appreciate your recognition of our efforts to address your suggestions, and we’re grateful for your decision to raise the score. Your support and constructive comments are very encouraging and helpful in improving our work.
>
> We also want to kindly remind you to remember to raise your score, as it is quite valuable and important to us. Thanks again for your time and support!

---

### Official Review · Reviewer_WDJh · 2025-07-03

**Clarity:** 2
**Significance:** 2
**Originality:** 2
**Rating:** 4
**Confidence:** 3

**Summary:**

This paper introduces CoMEM, a plug-and-play continuous memory mechanism for frozen vision-language models that encodes retrieved multimodal context into compact embeddings to bypass token overload. The core method utilizes a lightweight Q-Former to compress image–text retrievals into just eight dense vectors that are prepended to the model’s input. The proposed approach is highly efficient, requiring only 15.6k self-synthesized QA samples and LoRA fine-tuning of 1.2% of parameters while 6–12% improvements on multilingual VQA benchmarks.

**Questions:**

My main concerns are in the method section. I hope the author could provide more method insights and motivation. See the weakness above.

**Ethical Concerns:**

["NO or VERY MINOR ethics concerns only"]

**Final Justification:**

Thanks for authors' detailed and timely rebuttal. All of my questions have been answered with good answers and extensive qualitative results, and I am happy to raise my score.

**Limitations:**

The limitation discussions are sound and clear to me.

**Quality:**

2

**Strengths And Weaknesses:**

Strengths:
1. The idea of using the frozen VLM itself as a continuous memory encoder is simple and effective.
2. It achieves performance improvements with only 1.2% of parameters fine-tuned and 15.6k training data.
3. The evaluation is broad, spanning six English and two multilingual VQA benchmarks.

Weaknesses:
1. The paper lacks clarity in its writing, with many terms left unexplained. As a result, it is difficult to follow and understand. For example, Figure 1 does not clearly explain what the attention-based compression rates refer to.
2. The methodological novelty over prior Q-Former and RAG frameworks is limited and incremental.
3. The comparison to RAG seems unfair, as many recent studies have significantly improved its performance without substantial cost. I believe the evaluation should include the latest works in RAG, rather than relying solely on a vanilla baseline.
4. The trade-off between RAG and continuous memory is that RAG can be used in a training-free paradigm, whereas continuous memory typically requires training to adapt to the inference-time VLM which is not properly mentioned in the paper.
5. The paper lacks an ablation study to justify why using 8 embeddings is sufficient for the memory.
6. Inference latency of the whole framework is not evaluated.

---

> ### Author Rebuttal · Authors · 2025-07-31
>
> ## Rebuttal - Part 1
> ### **Weakness-1: Lacks Clarity in its Writing**
> > The paper lacks clarity in its writing, with many terms left unexplained. For example, Figure 1 does not clearly explain what the attention-based compression rates refer to.
>
> **Reply-to-Weakness1**:
>
> We appreciate the reviewer’s feedback and will add further clarifications to improve the readability of the paper in all figures and tables.
>
> First of all, we define the compression rate as:
> $$
> \\mathbf{Compression\\ Rate} = \\frac{\\#\\ \\text{tokens after compression}} {\\#\\ \\text{tokens before compression}} \\times 100\\%
> $$
> Here, we provide details for the experiment in Figure 1.
>   - The continuous memory module first encodes each retrieved item into continuous embeddings. We then compute the average attention scores across all layers within the VLM. Embeddings with the highest attention scores are retained with a compression rate, and the rest are discarded (same as described in Section 2.1, Memory Methods - VLM-as-Memory+Attn).
>   - Figure 1 demonstrates that, even with this simple token‑selection strategy, VLM‑as‑Memory methods outperform the baseline at high compression rates. This figure provides the intuition and preliminary evidence for our proposed method; therefore, we kept the explanation brief in the original submission.
>
>   To further clarify and prevent any possible ambiguities, we provide the following additional explanations:
>   - **Table 1** (in paper): This table compares various training‑free memory methods across three benchmarks (InfoSeek, OK‑VQA, A‑OKVQA). Methods include vanilla RAG, token pruning (FastV), and our VLM‑as‑Memory variants. The table shows that naive RAG can degrade performance due to context length overload, while our VLM‑as‑Memory methods consistently outperform token‑based methods because continuous embeddings align more naturally with the VLM.
>   - **Figure 3** (in paper): This figure analyzes our model’s long‑context robustness. We vary the number of retrieved image‑text pairs (x‑axis) and plot accuracy (y‑axis) for RAG and our method on InfoSeek. RAG performance drops when more than 30 items are retrieved because the increased context length introduces attention dilution and ambiguity. In contrast, our method remains stable across retrieval sizes since the compressed continuous memory does not excessively increase input length, demonstrating better scalability for long‑context scenarios.
>
> ### **Weakness-2: Novelty of the framework**
> > The methodological novelty over prior Q-Former and RAG frameworks is limited and incremental.
>
> **Reply-to-Weakness2**:
>
> In this paper, our target is to validate the effectiveness of using continuous memory for augmenting VLMs. Concretely, Q-Former is the technical way for achieving this goal, while RAG is a typical scenario for evaluating our approach. Follows are more detailed discussion about our contribution.
>   - **Our key contribution is to construct an information-rich and plug-and-play continuous memory with minimal cost.** To achieve this goal, we try to use simple and easy-to-follow techniques to verify whether the continuous memory is effective for improving VLMs. According to our extensive experiments, we validate that we can compress each multimodal item into just 8 continuous embeddings via the VLM itself, and the compressed embeddings are more helpful than SOTA RAG methods to better improve the performance.
>   - **Our design is meaningful and opens a new direction with many possibilities:**
>     - Firstly, it turns a VLM into its own memory encoder and achieves strong semantic alignment without modifying the base model.
>     - Secondly, our approach supports both multimodal and multilingual reasoning, and has shown strong performance across a broad range of tasks—including knowledge-intensive VQA, image captioning, and reasoning tasks—demonstrating its generality.
> | Method      | Knowledge-Intensive VQA (Average)  | Image Captioning (COCO) | Reasoning (MMMU) |
> |-------------|-------|------------------|-----------|
> | Qwen2.5-VL  | 30.8  | 0.39             | 32.22      |
> | +RAG        | 27.6  | 0.43            | 31.48      |
> | +CoMEM      | **35.4** | **0.46**      | **37.41**  |
>     - Thirdly, our memory also acts as a bridge between VLMs and LLMs, enabling the transfer of visual knowledge to pure language models.
> | LLM                        | InfoSeek: Unseen-Q | InfoSeek: Unseen-E | InfoSeek: All | OVEN: Query | OVEN: Entity | OVEN: All | Avg. |
> |---------------------------|--------------------|---------------------|----------------|--------------|---------------|-------------|--------------|
> | Qwen2.5-Instruct          | 5.0                | 4.8                 | 4.9            | 2.4          | 0.1           | 1.3         | 3.1          |
> | Qwen2.5-Instruct + RAG    | 13.4               | 10.3                | 11.9           | 1.8          | 2.7           | 2.2         | 7.0          |
> | Qwen2.5-Instruct + CoMEM  | **29.3**           | **27.4**            | **28.3**       | **6.8**      | **7.7**       | **7.2**     | **17.8**     |
>     - Importantly, our method opens up new possibilities such as grounding LLMs with vision, storing and accessing large-scale memory efficiently, and supporting long-horizon reasoning and planning in agent-based scenarios.
>   - **Our simple and effective designs well prove our concept, and also reduce the cost**: It has low training and inference cost with small inference latency, and the encoded continuous memory can be reused across tasks. We achieve strong performance using only 15.6k self-synthesized training samples and fine-tuning just 1.2% of parameters, making it both data and compute efficient, especially suitable for low-resource or cross-lingual settings. Across benchmarks and scenarios (knowledge-intensive VQA, image captioning, and reasoning tasks), our method consistently outperforms original vanilla RAG, up-to-date RAG baselines, confirming the effectiveness, robustness, and generalizability of our method.
>
> ### **Weakness-3 Unfair Comparison to RAG**
> > The evaluation should include the latest works in RAG, rather than relying solely on a vanilla baseline.
>
> **Reply-to-Weakness3**:
>
> Thank you for your suggestion. We agree that stronger and more diverse baselines are important for a fair evaluation, and we have selected SOTA ones in our paper for comparison.
>   - **Competitive RAG methods**: In Table 3 (in paper), we have compared our method with four highly relevant and latest multimodal RAG-based methods: Wiki-LLaVA, RORA-VLM, EchoSight, and ReflectiVA.
>   - **Adding More Experiments on RAG Baselines**: To address your concern, we have: (1) Re-implemented EchoSight and ReflectiVA (the only open-sourced methods among the four) and tested them on all benchmarks (MRAG, OKVQA, AOKVQA, ViQUAE) to provide a comprehensive evaluation. (2) Included more relevant RAG-based methods for comparison:
>     - **Wiki-LLaVA**[1]: Uses hierarchical retrieval from Wikipedia to add external knowledge for visual question answering.
>     - **RORA-VLM**[2]: Enhances robustness with two-stage retrieval and adversarial training to filter irrelevant content.
>     - **EchoSight**[3]: Applies a visual-first retrieval followed by text-image reranking for better encyclopedic knowledge integration.
>     - **ReflectiVA**[4]: Uses reflective tokens to decide when and what external knowledge to retrieve.
>     - **REVEAL**[5]: Pre-trains a retrieval-augmented VLM with a large multimodal memory for knowledge-intensive tasks.
>
>     The results are shown in the following Table 1. This extended evaluation demonstrates that CoMEM consistently outperforms open-source multimodal RAG methods (Echosight, ReflectiVA) across tasks, and is competitive with or superior to closed-source systems like RORA-VLM, REVEAL, etc. These results further validate the generality, scalability, and effectiveness of our proposed memory-based method.
>
> **Table 1: Performance Comparison Across Datasets**
>
> | Model              | infoseek(Q) | infoseek(E) | OVEN  | MRAG  | OKVQA | AOKVQA | ViQUAE |
> |--------------------|-------------|-------------|--------|--------|--------|---------|--------|
> | Wiki-LLaVA         | 28.6        | 25.7        | –      | –      | –      | –       | –      |
> | RORA-VLM           | 27.3        | 25.1        | 15.5   | –      | –      | –       | –      |
> | REVEAL             | –           | –           | –      | –      | 59.1   | 52.2    | –      |
> | Echosight          | 18.0        | 19.8        | 27.0   | 41.3   | 20.0   | 16.9    | 25.2   |
> | ReflectiVA         | 28.6        | 28.1        | 20.6   | 39.7   | 47.5   | 47.6    | 29.8   |
> | Qwen2-VL+CoMEM        | 32.6        | 33.1        | 23.6   | 35.1   | 57.7   | 60.6    | 36.3   |
> | Qwen2.5-VL+CoMEM      | 32.8        | 28.5        | 20.8   | 38.1   | 47.6   | 55.0    | 34.7   |
>
> [1] Davide Caffagni et al. Wiki-LLaVA: Hierarchical Retrieval-Augmented Generation for Multimodal LLMs. In CVPR 2024.
>
> [2] Jingyuan Qi et al. RoRA-VLM: Robust Retrieval-Augmented Vision Language Models. In 2024.
>
> [3] Yibin Yan . EchoSight: Advancing Visual-Language Models with Wiki Knowledge. In EMNLP 2024.
>
> [4] Federico Cocchi et al. Augmenting Multimodal LLMs with Self-Reflective Tokens for Knowledge-based Visual Question Answering. In CVPR 2025.
>
> [5] Ziniu Hu et al. REVEAL: Retrieval-Augmented Visual-Language Pre-Training with Multi-Source Multimodal Knowledge Memory. In CVPR 2023.

---

> ### Author Response · Authors · 2025-08-05
>
> Dear Reviewer,
>
> Thank you again for your initial review. We have carefully addressed your concerns in our rebuttal and provided detailed explanations along with additional experimental results to support our claims.
>
> We would greatly appreciate it if you could review our response and share your feedback. If there are any remaining concerns that require further clarification, we would be more than happy to provide more details and have a further discussion.
>
> We also wanted to note that the other reviewers have responded positively to our rebuttal and have updated their scores accordingly. We sincerely hope to receive your feedback as well before the discussion period ends.
>
> Best regards,
> The authors of Paper *Towards General Continuous Memory for Vision-Language Models*

---

> ### Author Response · Authors · 2025-08-05
>
> Dear Reviewer,
>
> During the rebuttal phase, we initially provided part of our response in the main rebuttal section, and included additional details in the *Author AC Confidential Comment* section, which we now realize is not visible to reviewers. To ensure you have access to the full context of our response, we are appending the missing portion below.
>
> ### **Weakness-4: Trade-off between RAG and Continuous Memory**
> > RAG can be used in a training-free paradigm, whereas continuous memory typically requires training to adapt to the inference-time VLM.
>
> **Reply-to-Weakness4**:
>
> Thanks for your suggestion. We agree that continuous memory requires training to adapt to the inference‑time VLM. However, a key advantage of our methods is that it significantly minimize both adaptation cost and training overhead:
>
> - **One-time training with cross-task generalization**: Unlike task‑specific fine‑tuning approaches, our framework requires only a single, lightweight training of the Q‑Former and LoRA adapters, which then generalizes across all downstream benchmarks. This design removes the need for retraining when switching tasks, thereby improving scalability and practical usability in multi‑task scenarios.
> - **Minimal adaptation cost**: We explicitly designed the framework for parameter‑efficient adaptation. Only ~1.2% of parameters are trained, and the base VLM remains frozen. This one‑time training cost is significantly lower than most SOTA RAG models that require larger adaptation modules or resource‑intensive fine‑tuning.
>
> ### **Weakness-5: Need an Ablation Study**
> > The paper lacks an ablation study to justify why using 8 embeddings is sufficient for the memory.
>
> **Reply-to-Weakness5**:
>
> Thank you for the suggestion. We conducted an ablation on the number of memory embeddings (4, 8, 16, 24) using the InfoSeek benchmark. As shown in Table 2, 8 embeddings consistently yield the best performance across both unseen question and unseen entity settings.
>
> - 4 embeddings underperform due to limited capacity for encoding complex multimodal and multilingual knowledge.
> - 16/24 embeddings introduce redundancy and noise, making training less efficient and harder to align with the frozen inference model.
> - 8 embeddings offer the best trade-off between expressiveness and compactness—sufficiently powerful, yet lightweight and efficient to train.
>
> These results suggest that 8 embeddings represent an effective inflection point, supporting our design choice in CoMEM and emphasizing the importance of tuning memory size for both performance and training efficiency.
>
> **Table 2: Performance Across Different Embedding Sizes on Infoseek**
>
> |  | 4-Embedding | 8-Embedding | 16-Embedding | 24-Embedding |
> |----|---|---|---|----|
> | Unseen Question Score   | 29.32       | 32.80        | 31.95        | 31.37        |
> | Unseen Entity Score     | 29.64       | 32.33       | 30.03        | 30.83        |
> | Final Score             | 29.48       | 32.74       | 30.96        | 31.10         |
>
> ### **Weakness-6: Latency**
> > Inference latency of the whole framework is not evaluated.
>
> **Reply-to-Weakness6**:
>
> We appreciate the reviewer’s suggestion and test the latency of baselines and our method. Concretely, we evaluated the throughput (tokens per second) for each model on 100 random sampled instances. The higher throughput indicates faster inference speed. All experiments were conducted on a single NVIDIA H100 (80 GB), and the results are summarized in Table 3 below:
> - The findings show that CoMEM maintains competitive inference speed as the original model, because it does not significantly increase input length. For RAG, as it expands the input by up to 15× and causes a proportional increase in attention cost, the inference speed decreases a lot. Since our CoMEM adds < 100 continuous memory tokens, this keeps inference cost close to that of the base model while improving accuracy.
> - Additionally, our training cost is the smallest among state‑of‑the‑art RAG models (Table 2 in paper). We train only lightweight LoRA adapters and the Q‑Former module (~1.2% of total parameters), while keeping the base VLM frozen. This parameter‑efficient design minimizes both compute and memory requirements during training.
>
> **Table 3: Latency**
>
> | Token per Second | InfoSeek | OVEN   | MRAG   | OKVQA  | AOKVQA | ViQuAE |
> |----|----|---|---|---|---|-----|
> | **Qwen2 models**|    |    |    |   |       |        |
> | Qwen2-VL        | 55.93   | 100.91 | 39.54  | 45.81 | 53.38 | 99.53  |
> | Qwen2-VL+RAG       | 40.71   | 49.27 | 8.38 | 34.66  | 23.60 | 37.50  |
> | Qwen2-VL+COMEM     | 48.93    | 53.29  | 18.61 | 40.42 | 51.60 | 41.50 |
> | **Qwen2.5 models**|        |        |        |        |        |        |
> | Qwen2.5-VL      | 53.44   | 72.71  | 42.13  | 45.64  | 42.87  | 53.49  |
> | Qwen2.5-VL+RAG     | 41.17    | 42.04  | 15.89  | 32.22  | 27.93  | 39.61  |
> | Qwen2.5-VL+CoMEM   | 51.05    | 54.39  | 19.79  | 42.86  | 61.41  | 48.81  |

---

> ### Author Response · Authors · 2025-08-06
>
> Dear Reviewer,
>
> We truly appreciate the time and effort you’ve already dedicated to reviewing our work. During the rebuttal phase, we invested significant effort in running additional experiments and providing detailed explanations to directly address your concerns.
>
> All other reviewers have kindly responded with constructive feedback and positive updates. We sincerely hope to hear your thoughts as well — your feedback is very important to us. If our responses have resolved your concerns, we would be grateful if you could consider updating your score accordingly. If there are any remaining questions or points you'd like us to clarify, we would be more than happy to continue the discussion.
>
> Thank you again for your time and consideration.
>
> Best regards,
> The authors of Paper *Towards General Continuous Memory for Vision-Language Models*

---

> ### Comment · Reviewer_WDJh · 2025-08-08
>
> Thanks for authors' detailed and timely rebuttal. All of my questions have been answered with good answers and extensive qualitative results, and I am happy to raise my score.

---

> > ### Author Response · Authors · 2025-08-08
> >
> > Dear Reviewer,
> >
> > Thank you very much for your thoughtful follow-up and your recognition of our efforts to address your questions! We're also sincerely grateful for your decision to raise the score—your support and encouragement mean a great deal to us.
> >
> > Best regards,
> >
> > The authors of Paper *Towards General Continuous Memory for Vision-Language Models*

---

> > > ### Author Response · Authors · 2025-08-09
> > > **Thank you**
> > >
> > > Dear Reviewer WDJh,
> > >
> > > As the rebuttal period is coming to a close, we would like to sincerely thank you for your valuable feedback and thoughtful suggestions. We truly appreciate the time and effort you’ve taken to engage with our work—your input has been incredibly helpful in guiding us to improve the paper.
> > >
> > > Thank you again for your support!
> > >
> > > Sincerely,
> > > The Authors

---

### Official Review · Reviewer_hjMA · 2025-07-03

**Clarity:** 3
**Significance:** 4
**Originality:** 4
**Rating:** 5
**Confidence:** 3

**Summary:**

The paper proposes CoMEM, a continuous memory mechanism for vision-language models (VLMs), using the VLM itself to encode multimodal and multilingual knowledge into compact embeddings. It outperforms traditional RAG and token pruning in tasks like InfoSeek and OKVQA, enabling efficient, plug-and-play integration with strong performance in long-context and low-resource language settings.

**Questions:**

1. Why do continuous embeddings work better than discrete tokens?
The results show clear advantages but lack explanation. Is it because of higher compression efficiency or better compatibility with the VLM's attention mechanism? A simple theoretical analysis or comparison of different compression rates (e.g., 8-dim vs 16-dim embeddings) would make this more convincing.

1. Does this work only for QA tasks?
All current experiments are on question answering. Could you test it on generation tasks like image captioning? Even one small experiment on COCO captioning would prove the method's generalizability.

**Ethical Concerns:**

["NO or VERY MINOR ethics concerns only"]

**Final Justification:**

The rebuttal effectively addressed my concerns. The authors provided both intuitive explanations and ablation studies (e.g., varying memory size) to clarify the benefits of continuous embeddings over discrete tokens. They also extended their evaluation beyond QA tasks with a caption generation experiment on COCO, showing consistent improvements across several standard metrics.

These additions strengthen the paper's clarity, empirical support, and generalizability. Given the overall contribution, the practical value of the proposed mechanism, and the quality of the response, I have raised my rating from 4 to 5.

**Limitations:**

Yes

**Paper Formatting Concerns:**

Line 122: "5This"

**Quality:**

4

**Strengths And Weaknesses:**

## Strengths
- **Strong Empirical Results**: Demonstrates consistent improvements across eight multimodal benchmarks, with significant gains over baselines.
- **Efficient Design**: Uses only 1.2% additional parameters and a small self-synthesized dataset, making it practical for deployment.
- **Novel Approach**: Introduces the idea of a VLM serving as its own memory encoder, avoiding costly alignment training and enabling plug-and-play integration.
- **Scalability**: Handles long-context inputs robustly, unlike RAG methods that degrade with more retrieved items.

## Weaknesses
- **Limited Theoretical Justification**: Lacks analysis of why continuous embeddings outperform discrete tokens or how compression affects performance.
- **Narrow Benchmarking**: Focuses heavily on VQA tasks; testing on broader tasks (e.g., captioning) would strengthen claims of generalizability.

---

> ### Author Rebuttal · Authors · 2025-07-30
>
> ## Rebuttal
>
> We sincerely thank the reviewer for the insightful comments and appreciate the positive feedback! We reply to all the concerns raised in the weaknesses and questions part.
> ### **Weakness-1: Limited Theoretical Justification**
> > Lacks analysis of why continuous embeddings outperform discrete tokens or how compression affects performance.
>
> **Reply-to-Weakness1-A**:
>
> For continuous embeddings and discrete tokens, it is hard to propose a proof that conclusively establishes the superiority of either approach. Instead, we demonstrate the intuitive and practical advantages of continuous embeddings from several perspectives.
> - **Conceptual alignment with human cognition**: Cognitive‑science research shows that humans store and manipulate knowledge at the level of abstract concepts, not at the level of exact pixel patterns or phoneme sequences. When you think of “the Great Wall” or “the Eiffel Tower,” you effortlessly summon a coarse, high‑level mental image while discarding countless low‑level details such as the precise arrangement of every brick or rivet. Therefore, the continuous embedding based memory better mirrors the way our brains compress and retrieve multimodal information.
> - **Continuous embeddings better support high compression**: as continuous embeddings can densely encode the information, which prevents the great increase of context length (see Table 2). It is rather helpful for VLMs to read and understand massive multimodal data. For the example of RAG scenario, here we list the context length in three settings: Original model(no retrieval) , Qwen2-VL+RAG and our CoMEM method.  We can clearly see that although we only retrieve 10 examples in the context, the context length increases by approximately **15×** when using RAG. It will cause the VLM to suffer from the long context understanding problem. In contrast, our method converts the retrieved examples into continuous embeddings, which leads to fewer than 100 additional embeddings to the input sequence. It avoids the long context understanding problem and also reduces the cost of encoding such long input sequences.
> - **Better empirical results of continuous embedding**: since continuous embeddings generally have higher compression rate, their rather short context length allows the model to process multimodal information more effectively. In our method, we jointly compress both image and text knowledge into a unified continuous latent space, which enables better cross‑modal integration and achieves superior performance on most benchmarks compared to state‑of‑the‑art methods. We also add results comparison with up-to-date RAG methods. We also added comparisons with up‑to‑date RAG baselines, and CoMEM consistently delivers the best results across all benchmarks:
>
> **Table 1: Performance Comparison Across Datasets**
>
> | Model              | infoseek(Q) | infoseek(E) | OVEN  | MRAG  | OKVQA | AOKVQA | ViQUAE |
> |--------------------|-------------|-------------|--------|--------|--------|---------|--------|
> | Wiki-LLaVA         | 28.6        | 25.7        | –      | –      | –      | –       | –      |
> | RORA-VLM           | 27.3        | 25.1        | 15.5   | –      | –      | –       | –      |
> | REVEAL             | –           | –           | –      | –      | 59.1   | 52.2    | –      |
> | Echosight          | 18.0        | 19.8        | 27.0   | 41.3   | 20.0   | 16.9    | 25.2   |
> | ReflectiVA         | 28.6        | 28.1        | 20.6   | 39.7   | 47.5   | 47.6    | 29.8   |
> | Qwen2+CoMEM        | 32.6        | 33.1        | 23.6   | 35.1   | 57.7   | 60.6    | 36.3   |
> | Qwen2.5+CoMEM      | 32.8        | 28.5        | 20.8   | 38.1   | 47.6   | 55.0    | 34.7   |
>
>
> **Table 2: Token Number and Accuracy Across Datasets**
> | **Token Number (Accuracy)** | **Infoseek**       | **Oven**          | **MRAG**          | **OKVQA**         | **AOKVQA**        | **ViQuAE**        |
> |-----------------------------|--------------------|-------------------|-------------------|-------------------|-------------------|-------------------|
> | **Qwen2-VL**                | 357.6 (17.8)      | 408.6 (25.5)     | 84.5 (39.3)      | 404.8 (36.3)      | 418.4 (41.8)      | 424.7 (34.5)      |
> | **Qwen2-VL+RAG**            | 6163.8 (19.0)     | 6481.5 (24.7)    | 5882.4 (40.4)    | 6803.8 (41.9)     | 6715.4 (45.3)     | 6943.2 (33.6)     |
> | **Qwen2-VL+COMEM**          | 416.8 (33.1)       | 492.9 (30.5)      | 176.9 (35.1)      | 453.8 (57.7)      | 478.3 (60.6)      | 489.2 (36.3)     |
>
> **Reply-to-Weakness1-B**:
>
> Comparison of different compression rates: We conducted an ablation study by varying the number of memory embeddings (4, 8, 16, 24) on the InfoSeek benchmark, to study the effectiveness of different compression rates. As shown in Table 3 below, 8 embeddings consistently achieve the best overall performance across both unseen question and unseen entity settings.
>   - 4 embeddings do not perform well due to limited capacity—it is difficult to encode complex multimodal and multilingual knowledge with such a compact representation.
>   - 16 and 24 embeddings perform slightly worse than 8. While they offer more representational capacity, they also introduce redundancy and noise, making them harder to train effectively. These longer embeddings require more data to properly align with the frozen inference model, reducing training efficiency.
>   - 8 embeddings strike an optimal balance between compactness and expressiveness. They are sufficient to capture relevant knowledge while remaining lightweight, easy to integrate, and efficient to train.
>
> These results suggest that 8 embeddings represent an effective inflection point, supporting our design choice in CoMEM and emphasizing the importance of tuning memory size for both performance and training efficiency.
>
> **Table 3: Performance Across Different Embedding Sizes on Infoseek**
> |                         | 4-Embedding | 8-Embedding | 16-Embedding | 24-Embedding |
> |-------------------------|-------------|-------------|--------------|--------------|
> | Unseen Question Score   | 29.32       | 32.80        | 31.95        | 31.37        |
> | Unseen Entity Score     | 29.64       | 32.33       | 30.03        | 30.83        |
> | Final Score             | 29.48       | 32.74       | 30.96        | 31.10         |
>
> ### **Weakness-2: Narrow Benchmarking**
> > Focuses heavily on VQA tasks; testing on broader tasks (e.g., captioning) would strengthen claims of generalizability.
>
> **Reply-to-Weakness2**:
>
> Following the suggestion of the reviewer, we evaluate the generalizability of CoMEM on a caption generation task using the COCO 2014 dataset. We randomly sampled **100 image-caption pairs** from the test set and used CLIP to retrieve relevant image-caption pairs from the training set. We then compared three setups: (1) Original model (no retrieval); (2) RAG-style retrieval (retrieved image-caption pairs prepended in prompt); (3) Our method (retrieved image-caption pairs encoded into continuous memory). We tested both Qwen2.5 and Qwen2, and report standard captioning metrics below in Table 4. We obtain the following conclusions:
>   - **Better precision and fluency**: BLEU-1 / BLEU-4 measure n-gram precision; METEOR balances precision and recall with synonym matching; higher scores indicate better word accuracy. CoMEM consistently improves BLEU-1, BLEU-4, and METEOR across both models, indicating more fluent and accurate captions.
>   - **Substantial gain in content relevance**: CIDEr scores use tf-idf weighted n-grams to measure relevance to ground-truth captions. It's increased significantly in Qwen2.5-VL with CoMEM (from 0.24 → 0.64), showing better content relevance to ground-truth captions.
>   - **Improved semantic similarity**: ROUGE-L captures the longest common subsequence with the reference, reflecting surface-level overlap. BERTScore-F computes semantic similarity using contextual embeddings. Memory-augmented models generate captions with high ROUGE-L and BERTScore-F scores, showing stronger lexical overlap and semantic similarity.
>
> These results confirm that CoMEM generalizes well beyond QA tasks, offering an effective and lightweight memory mechanism for open-ended generation tasks like image captioning.
>
> **Table 4: Evaluation Metrics for Qwen2.5-VL and Qwen2-VL on COCO Captioning task**
> | Model | Method  | BLEU-1 | BLEU-4 | METEOR | CIDEr | ROUGE-L | BERTScore-F | Trigram Diversity |
> |------|-------|------|------|--------|--------|----------|--------------|--------------------|
> | qwen2.5-VL    | Original| 0.26   | 0.04   | 0.18   | 0.24   | 0.26     | 0.81         | 0.92               |
> |            | +RAG     | 0.30   | 0.05   | 0.20   | 0.45   | 0.27     | 0.81         | **0.94**           |
> |            | +CoMEM  | **0.34** | **0.07** | **0.21** | **0.64** | **0.32**   | 0.81         | 0.86               |
> | qwen2-VL      | Original| 0.34   | 0.08   | 0.21   | 0.79   | 0.34     | 0.81         | 0.84               |
> |            | +RAG     | 0.34   | 0.07   | 0.20   | 0.64   | 0.32     | 0.81         | 0.83               |
> |            | +CoMEM  | **0.36** | 0.08   | **0.23** | 0.76   | 0.32     | **0.82**     | **0.88**           |
>
> ### **Question-1**
> > Why do continuous embeddings work better than discrete tokens? The results show clear advantages but lack explanation. Is it because of higher compression efficiency or better compatibility with the VLM's attention mechanism? A simple theoretical analysis or comparison of different compression rates (e.g., 8-dim vs 16-dim embeddings)  would make this more convincing.
>
> **Reply-to-Question-1**:
>
> Please refer to Reply-to-Weakness1-A and Reply-to-Weakness1-B
> ### **Question-2**
> > Does this work only for QA tasks? All current experiments are on question answering. Could you test it on generation tasks like image captioning? Even one small experiment on COCO captioning would prove the method's generalizability.
>
> **Reply-to-Question-2**:
>
> Please refer to Reply-to-Weakness2

---

> > ### Comment · Reviewer_hjMA · 2025-08-03
> > **Acknowledgement of rebuttal**
> >
> > The rebuttal addresses my main concerns. The authors provide additional justification and ablations (e.g., varying memory size) for the advantages of continuous embeddings, which strengthen the rationale behind the design choice. They also extend evaluation beyond QA tasks with a COCO captioning experiment, showing consistent improvements across multiple metrics. These additions improve both the clarity and the demonstrated generality of the approach. I will take these clarifications into account and revise my evaluation accordingly.

---

> > > ### Author Response · Authors · 2025-08-04
> > >
> > > We sincerely thank the reviewer for the thoughtful and constructive feedback! We're glad that the additional performance comparison, justifications for continuous embeddings, and the COCO captioning experiment helped clarify and strengthen the rationale and generality of our approach.
> > >
> > > We truly appreciate your willingness to take these clarifications into account when revising your evaluation. If there are any remaining questions you'd like us to elaborate on, we would be more than happy to provide more details and engage in further discussion. We greatly value your feedback and hope that the improvements can be reflected in your final score.

---

### Author Response · Authors · 2025-08-07

Dear Reviewers,

We would like to sincerely thank all of you for your valuable time, thoughtful feedback, and constructive discussions! We truly appreciate the depth and clarity of your comments, which have been instrumental in helping us improve our work. The feedback we received was insightful and encouraging, and it has played a crucial role in shaping the current version of our paper.

In response to your suggestions, In response, we conducted a wide range of additional experiments and provided detailed clarifications to strengthen our submission. These include:

---
###  Additional Experiments

- Method Robustness

    - **Experiment 1: Ablation Study on Embedding Size**: We varied the number of memory embeddings (4, 8, 16, 24) and found that 8 embeddings strike the optimal balance between compactness and performance.

    - **Experiment 2: Extended Comparison with Strong RAG Baselines**: We benchmarked against advanced RAG methods (Wiki-LLaVA, RORA-VLM, REVEAL, EchoSight, ReflectVA) and showed that CoMEM consistently outperforms them.

    - **Experiment 3: Transferability Across VLMs and LLMs**: We applied CoMEM to Qwen2.5-Instruct (a text-only LLM) and demonstrated strong transferability and plug-and-play compatibility across model types.

- Method Generalizability

    - **Experiment 4: Evaluation on Image Captioning (COCO Dataset)**: We tested on the COCO captioning task and observed significant improvements in fluency, precision, and semantic similarity across all metrics.

    - **Experiment 5: Evaluation on Reasoning Tasks (MMMU & MathVista)**: We evaluated CoMEM on reasoning-heavy benchmarks and confirmed robust gains in accuracy and similarity across diverse domains.

- Method Cost

    - **Experiment 6: Inference Latency Test**: We measured inference speed and showed that CoMEM maintains near-original throughput, unlike RAG which incurs significant slowdown.

    - **Experiment 7: Token Cost Comparison**: We compared token usage across Original, RAG, and CoMEM pipelines, showing CoMEM reduces context length by ~15× while preserving performance.

###  Clarifications and Improvements

- **Clarification 1: Theoretical Motivation for Continuous Embeddings**: We articulated the cognitive and efficiency-based rationale for using continuous embeddings over discrete tokens.

- **Clarification 2: Compression Rate Definition**: We added a formal definition of compression rate to clarify how compression selection affects input length.

- **Clarification 3: Memory Mechanism Explanation**: We explained that CoMEM functions as a lightweight and general memory encoder, converting retrieved knowledge into dense embeddings for direct input into VLMs.

- **Clarification 4: Inference Process Explanation**: We detailed how memory embeddings are generated and integrated during inference without modifying the base model.

- **Clarification 5: Improved Writing and Figure Clarity**: We revised ambiguous figures and added missing definitions to enhance readability and understanding.

- **Clarification 6: Motivation Behind RAG Limitations**: We provided empirical evidence showing how long-input sequences in RAG degrade performance, reinforcing the benefits of compression in CoMEM.

---

We hope these additions clearly address the concerns raised and further demonstrate the effectiveness, efficiency, and generalizability of our proposed method.

We will integrate the results of these new experiments into the revised version of the paper and refine the writing to make the explanations clearer and more precise.

We are also grateful that the majority of reviewers (**3 out of 4**) provided timely and positive feedback during the rebuttal phase. We are pleased to know that our additional experiments and clarifications have addressed your concerns, and we truly appreciate your decision to raise the scores accordingly.

We remain more than happy to continue the discussion if needed, and we sincerely hope that the remaining reviewer can also provide feedback. Your input is very important to us and would be greatly appreciated.

Thank you again for your thoughtful reviews and constructive engagement.


Best regards,

The authors of Paper *Towards General Continuous Memory for Vision-Language Models*

---

### Note · Authors · 2025-08-11

Dear NeurIPS 2025 AC, SAC, and PC,

We sincerely thank you and the reviewers for your time and feedback. During the rebuttal phase, we made substantial efforts to address all reviewer concerns through extensive new experiments and detailed clarifications, which we believe significantly strengthened our submission.

Firstly, we would like to reiterate our core contributions, which have been recognized by the reviewers:

- **Novelty**: CoMEM introduces a plug-and-play continuous memory mechanism, enabling the VLM itself to serve as a memory encoder.
- **Efficiency**: The method is highly efficient, requiring only 1.2% of parameters and a small self-synthesized dataset.
- **Strong empirical results**: Demonstrated consistent improvements across eight benchmarks, including long-context and multilingual tasks.
- **Scalability and broad applicability**: CoMEM is scalable, broadly applicable, and practical for real-world deployment scenarios.

Secondly, the main concerns raised by reviewers focused on:
- Generalization: Applicability of CoMEM to tasks beyond VQA, such as captioning and reasoning.
- Comparative Evaluation: Performance against recent strong RAG-based baselines.
- Method Understanding: Justification for using 8 embeddings and the advantages of continuous memory.

In response, we conducted seven new experiments to demonstrate the robustness, generalizability, and efficiency of CoMEM:

- **Ablation on memory size**: Confirming the optimal balance between compactness and performance.
- **Comparison with strong RAG baselines** (e.g., Wiki-LLaVA, REVEAL, RORA-VLM): Showing consistently strong and robust outperformance.
- **Generalization to captioning and reasoning tasks**: Demonstrating significant gains on COCO, MMMU, and MathVista.
- **Efficiency evaluation**: Showing CoMEM reduces context length by ~15× with minimal latency overhead.
- **Transferability to other models**: Applying CoMEM to the text-only Qwen2.5-Instruct, confirming plug-and-play compatibility across model types.

We also provided key clarifications, including theoretical motivations, implementation details, and improved figure clarity. All of these enhancements will be incorporated into the final version of the paper.

We truly appreciate the opportunity to improve our paper through the review process. We hope these substantial additions, together with the core strengths of our method, will support a favorable decision.

Thank you again for your time and consideration.

The Authors

---

### Decision · Program_Chairs · 2025-09-17

**Decision:**

Accept (poster)

**Comment:**

The paper received all positive reviews, leading to a final acceptance recommendation.